# Examining the human infectious reservoir for *Plasmodium falciparum* malaria in areas of differing transmission intensity

Bronner P. Gonçalves[1], Melissa C. Kapulu[2,3], Patrick Sawa[4], Wamdaogo M. Guelbéogo[5], Alfred B. Tiono[5], Lynn Grignard[1], Will Stone[1,6], Joel Hellewell[7], Kjerstin Lanke[6], Guido J.H. Bastiaens[6], John Bradley[8], Issa Nébié[5], Joyce M. Ngoi[2], Robin Oriango[4], Dora Mkabili[2], Maureen Nyaurah[4], Janet Midega[2,9], Dyann F. Wirth[10], Kevin Marsh[2,3], Thomas S. Churcher ⓘ [7], Philip Bejon[2,3], Sodiomon B. Sirima[5], Chris Drakeley[1] & Teun Bousema[1,6]

A detailed understanding of the human infectious reservoir is essential for improving malaria transmission-reducing interventions. Here we report a multi-regional assessment of population-wide malaria transmission potential based on 1209 mosquito feeding assays in endemic areas of Burkina Faso and Kenya. Across both sites, we identified 39 infectious individuals. In high endemicity settings, infectious individuals were identifiable by research-grade microscopy (92.6%; 25/27), whilst one of three infectious individuals in the lowest endemicity setting was detected by molecular techniques alone. The percentages of infected mosquitoes in the different surveys ranged from 0.05 (4/7716) to 1.6% (121/7749), and correlate positively with transmission intensity. We also estimated exposure to malaria vectors through genetic matching of blood from 1094 wild-caught bloodfed mosquitoes with that of humans resident in the same houses. Although adults transmitted fewer parasites to mosquitoes than children, they received more mosquito bites, thus balancing their contribution to the infectious reservoir.

[1] Department of Immunology and Infection, London School of Hygiene & Tropical Medicine, London WC1E 7HT, UK. [2] Kenya Medical Research Institute (KEMRI)-Wellcome Trust Programme, PO Box 230 , Kilifi 80108, Kenya. [3] Centre for Tropical Medicine and Global Health, Nuffield Department of Clinical Medicine, University of Oxford, Oxford OX3 7FZ, UK. [4] Human Health Division, International Centre of Insect Physiology and Ecology, PO Box 30Mbita Point, Western Kenya 40305, Kenya. [5] Department of Biomedical Sciences, Centre National de Recherche et de Formation sur le Paludisme, Ouagadougou 01 BP 2208, Burkina Faso. [6] Radboud Institute for Health Sciences, Radboud University Medical Center, 6525 GA Nijmegen The Netherlands. [7] MRC Centre for Outbreak Analysis & Modelling, Department of Infectious Disease Epidemiology, Imperial College London, London W2 1PG, UK. [8] MRC Tropical Epidemiology Group, Department of Infectious Disease Epidemiology, London School of Hygiene & Tropical Medicine, London WC1E 7HT, UK. [9] Centre for Genomics and Global Health, Wellcome Trust Centre for Human Genetics, University of Oxford, Oxford OX3 7BN, UK. [10] Department of Immunology and Infectious Diseases, Harvard T.H. Chan School of Public Health, Boston, MA 02115, USA. Bronner P. Gonçalves and Melissa C. Kapulu contributed equally to this work. Correspondence and requests for materials should be addressed to C.D. (email: Chris.Drakeley@lshtm.ac.uk) or to T.B. (email: Teun.Bousema@radboudumc.nl)

Heterogeneity in the transmission potential of individual hosts is a common feature of many infectious diseases, and the identification of individuals who disproportionately contribute to onward transmission has attracted much attention[1, 2]. For vector-borne infections, quantifying the contribution of individual humans to transmission (the infectious reservoir) requires the estimation of host infectivity with the pathogen and assessment of the effective contact rate, the frequency with which individual hosts are sampled by the vector population. Malaria due to *Plasmodium falciparum* remains a major cause of morbidity and mortality worldwide, despite marked recent declines in disease burden[3, 4]. Initiatives to further reduce the burden of malaria, as well as efforts to contain the spread of artemisinin resistant malaria parasites in Southeast Asia[5], require a thorough understanding of the human infectious reservoir for malaria, which would allow interventions to be targeted to individuals who are most important for the transmission of infection to mosquitoes. The need to resolve uncertainties in identifying these individuals is further necessitated by accumulating evidence that a considerable proportion of those infected have very low parasite densities, only detectable by sensitive molecular assays. This need is particularly acute in low malaria transmission settings and areas threatened by artemisinin resistance[6, 7].

Few epidemiological studies have assessed malaria infectiousness at a population level by directly quantifying human infectivity using mosquito feeding experiments[8–12]. In these experiments, malaria vectors are fed directly on the skin or on blood through a membrane and later dissected for parasite development assessment, such as oocyst detection. Only one of these studies, conducted in an area of intense malaria transmission in Burkina Faso, concurrently quantified asexual stage parasites and the transmissible sexual stage parasites (gametocytes) by molecular assays and concluded that up to 17% of mosquito infections are caused by submicroscopic parasite carriage in humans[12]. In contrast, in Cambodia a hospital-based study involving uncomplicated malaria cases reported much lower infectivity of submicroscopic infections[13]. The findings of these studies highlight the need for more extensive assessments of infectivity to mosquitoes across a range of endemicities. Such xenodiagnostic surveys select individuals regardless of parasite carriage, incorporate concurrent assessments of parasite and gametocyte densities by molecular assays, which could provide insights into the mechanisms responsible for the variation in

human-to-mosquito infectiousness, and take into account that individuals are not equally bitten by mosquitoes and consequently have different numbers of opportunities to transmit[14]. Both infectiousness and exposure to mosquito bites are needed to be able to quantify an individual's contribution to malaria transmission.

In this study, we determined the proportions of mosquito infections originating from different demographic groups in areas with low, moderate and high malaria transmission intensity. For this, we integrated infectivity data from xenodiagnostic surveys, molecular parasite quantification and data on actual exposure to malaria vectors, which were generated by blood meal analysis of wild-caught bloodfed mosquitoes in study sites in East and West Africa. We observe that infectiousness is more prevalent in areas with high malaria transmission intensity compared to low-endemic regions, and that although children are often more infectious than adults, adults receive more mosquito bites than children, which amplifies their contribution to malaria infections in mosquitoes.

## Results

**Epidemiology of malaria parasites and gametocytes.** Xenodiagnostic surveys were performed in four areas selected to represent three different malaria endemicities (Table 1): in Burkina Faso, the villages of Laye and Balonghin, sampled during the low intensity season (from here on 'dry season') of 2013 and the peak intensity season (from here on 'wet season') of 2014 respectively, are both characterised by intense seasonal transmission; malaria transmission in Mbita, Kenya, is seasonal with moderate intensity; in Kilifi, also located in Kenya, low transmission occurs throughout the year but is higher during the wet season months. Membrane feeding experiments were used to quantify infectivity of study participants to locally reared *Anopheles gambiae* mosquitoes[12, 15].

A total of 1075 individuals were recruited regardless of their parasitological status, including 142 who participated in both dry and wet season surveys in Mbita (Table 1), with a total of 1216 parasitological observations. As expected, infection with *P. falciparum* was more prevalent in high (Burkina Faso) vs. moderate (Mbita) and low (Kilifi) transmission settings (Fig. 1a). While parasite positivity was higher during wet vs. dry season in Burkina Faso (83.8% [166/198] and 50.5% [100/198] by *18S* quantitative PCR (qPCR), respectively), in Kilifi, *P. falciparum*

| | Laye (Burkina Faso) | Balonghin (Burkina Faso) | Mbita (Kenya) | Mbita (Kenya) | Kilifi (Kenya) | Kilifi (Kenya) |
|---|---|---|---|---|---|---|
| **Table 1 Xenodiagnostic surveys.** | | | | | | |
| Transmission | | | | | | |
| Intensity | High | | Moderate | | Low | |
| Season | Dry | Wet | Dry | Wet | Dry | Wet |
| Number of participants | 200 | 200 | 202 | 202 | 139[a] | 274 |
| Start date | May-2013 | Oct-2014 | Feb-2014 | May-2014 | Jan-2014 | May-2014 |
| End date | Jun-2013 | Nov-2014 | Apr-2014 | Jul-2014 | Apr-2014 | Dec-2014 |
| | N (%) | N (%) | N (%) | N (%) | N (%) | N (%) |
| Age categories (in years) | | | | | | |
| <5 | 50 (25.0) | 49 (24.6) | 41 (20.3) | 50 (24.7) | 16 (11.5) | 66 (24.1) |
| 5–15 | 100 (50.0) | 100 (50.2) | 59 (29.2) | 51 (25.2) | 33 (23.7) | 76 (27.7) |
| >15 | 50 (25.0) | 50 (25.1) | 102 (50.5) | 101 (50.0) | 90 (64.7) | 132 (48.2) |
| Gender | | | | | | |
| Male | 98 (49.7) | 114 (57.3) | 69 (34.2) | 86 (42.6) | 43 (30.9) | 108 (39.4) |
| Female | 99 (50.2) | 85 (42.7) | 133 (65.8) | 116 (57.4) | 96 (69.1) | 166 (60.6) |

[a]35 Participants were recruited in January–February 2015

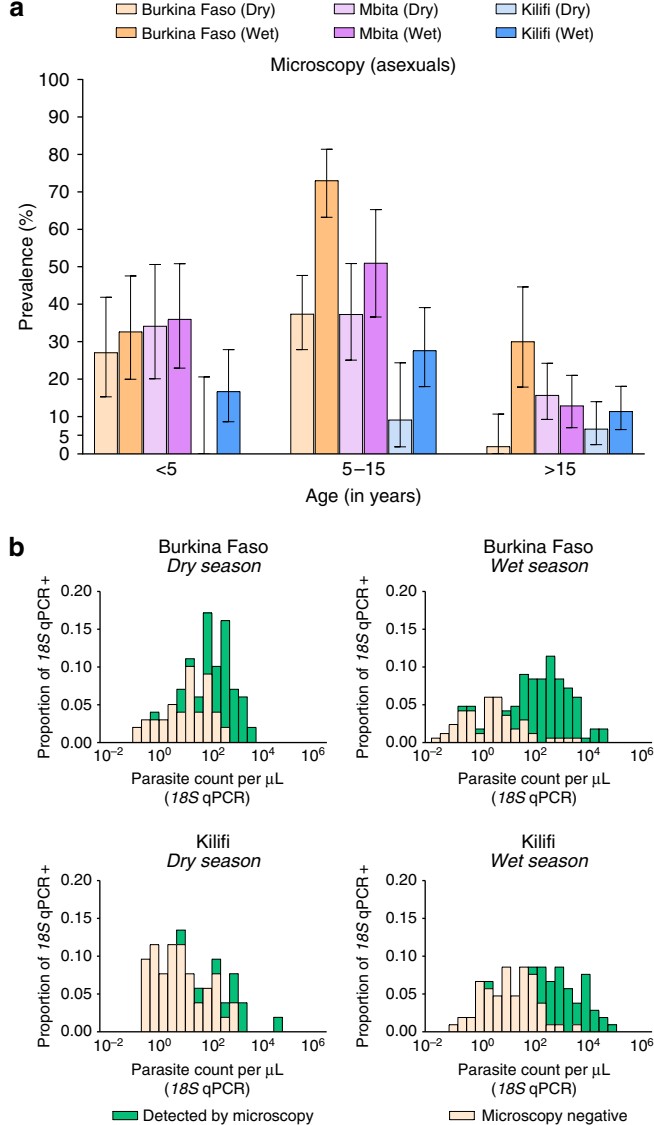

**Fig. 1** Age-specific asexual stage parasites prevalence by microscopy (**a**) and distributions of parasite densities (**b**) by study site. In **a**, 95% exact binomial confidence intervals are presented for microscopy-based parasite prevalence estimates. All participants, except three children in Laye and one in Balonghin, had microscopy results available; numbers of study subjects by age group and survey are presented in Table 1. In **b**, parasite densities quantified by DNA-based *18S* qPCR are presented (log₁₀ scale) for both patent and subpatent infections. This panel only includes *18S* qPCR-positive samples (100, 166, 52 and 105 in Burkina Faso dry and wet season surveys, and Kilifi dry and wet season surveys, respectively)

prevalence was similar in both seasons (37.4% [52/139] vs. 38.3% [105/274] by *18S* qPCR during dry and rainy seasons, respectively). Molecular detection and quantification of malaria parasites failed for samples collected in Mbita due to local electricity issues resulting in an unknown number and duration of freeze-thaw cycles. Based on microscopy results, the prevalences of asexual stage parasites in the two Mbita surveys were also similar (25.7% [52/202] vs. 28.2% [57/202] during dry and wet seasons, respectively). In all sites, children aged between 5 and 15 years had higher parasite prevalence than individuals older than 15 years (Fig. 1a). The detectability of infections by microscopy was associated with qPCR parasite density

(Supplementary Fig. 1); the median (interquartile range [IQR]) parasite densities estimated by *18S* qPCR in microscopically subpatent and patent infections were 4.9 (1.0–32.6) and 387.3 (95.1–1244.5) parasites per µL, respectively. In Balonghin, Laye, Kilifi wet season and Kilifi dry season the percentages of infections that were detectable by *18S* qPCR but negative by microscopy (i.e. submicroscopic infections) were 37.9%, 49.5%, 58.1% and 82.7%, respectively (Fig. 1b). The distributions of parasite densities estimated by *18S* qPCR suggest that the tails of undetectable infections (infections with densities below 0.01 parasites per µL) are negligible for all settings except in Kilifi dry season and may be interpreted as evidence that our assays detected the majority of malaria infections[6]. *18S* qPCR parasite densities were lower in adults compared to children living in the same study area (*P* < 0.001 in a multivariable negative binomial model that adjusted for season and study region).

A small percentage (6.2% [75/1213]) of samples had microscopically detectable levels of gametocytes. As expected, *Pfs25* mRNA quantitative nucleic acid sequence-based amplification (QT-NASBA) identified gametocytes in a much larger percentage of the study population (353/809 [43.6%] participants in Burkina Faso and Kilifi surveys). Overall, 76.6% (324/423) of all *18S* qPCR parasite positive individuals were also *Pfs25* mRNA QT-NASBA gametocyte positive. *Pfs25* mRNA QT-NASBA identified more gametocyte carriers in the wet compared to the dry season in both high-endemic Burkina Faso (75.2% [149/198] vs. 49.5% [98/198]) and low-endemic Kilifi (29.6% [81/274] vs. 18.0% [25/139]). Similarly, gametocyte densities estimated by *Pfs25* mRNA QT-NASBA were higher during wet compared to dry season in Kilifi and in Burkina Faso (*P* < 0.001 in a multivariable negative binomial model that adjusted for age and study region). As with asexual stage parasites, school children were more frequently gametocytaemic than adults (63.0 vs. 34.0% in dry-season Laye [*P* = 0.001], 86.9 vs. 78.0% in wet-season Balonghin [*P* = 0.16] and 44.7 vs. 26.5% in Kilifi during the wet season [*P* = 0.007], respectively; *P*-values by the χ² test), except in Kilifi during the dry season (12.1 vs. 21.1%, respectively [*P* = 0.26]).

**Exposure to mosquitoes is dependent on age and setting**. To generate realistic quantification of the transmission potential of individuals, we measured exposure to anopheline vectors to determine the likely frequency with which potentially infectious individuals are sampled by mosquitoes. We analysed a total of 1874 blood meals from fed mosquitoes (1066 from Balonghin and 808 from Mbita) collected in households in the study villages to estimate human host age-specific exposure to malaria vectors. Sampling bloodfed mosquitoes in the low endemic area in Kilifi was abandoned when 2 months of sampling yielded only 31 bloodfed mosquitoes, which was insufficient for meaningful assessments of mosquito exposure by age groups. Molecular typing of blood meals to identify the human blood source using a short tandem repeat multiplex PCR assay was successfully performed for 966 and 689 mosquitoes collected in Burkina Faso and Kenya, respectively. Blood was taken from 99.2% (126/127) and 86.7% (163/188) of all household members in Balonghin and in Mbita, respectively, to link to mosquito blood meals.

In total, 666 and 428 mosquito blood meals were matched to single individuals living in the same household where they were collected in Balonghin and Mbita, respectively. Since the association between mosquito exposure and age varied with study site (*P* = 0.001 in a mixed effects negative binomial model; Fig. 2a, c, d), we performed separate analyses for the two sites. In Burkina Faso, adults (incidence rate ratio (IRR) 20.9 95% confidence interval [CI] 7.7–57.4) and children aged 5–15 years (IRR 7.7 95% CI 2.9–20.8) were more often bitten by *Anopheles*

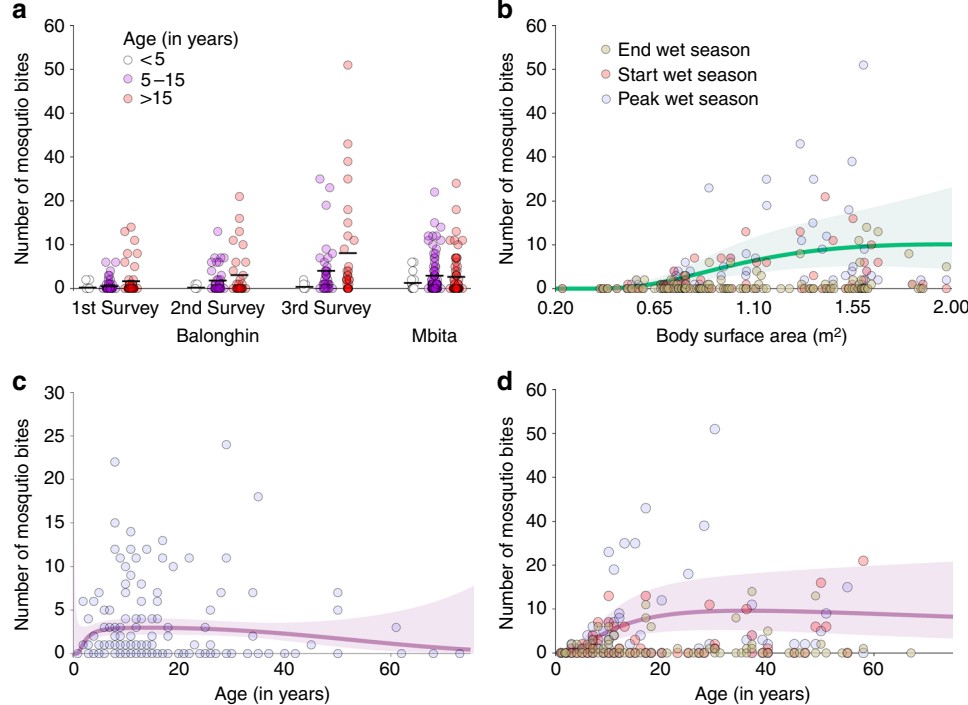

**Fig. 2** Mosquito exposure by age and body surface area. In **a**, the number of mosquito bites for each age group and survey is presented; each circle represents a study participant, and the mean number of mosquito bites per individual by age group is represented by horizontal black lines. In **b**–**d**, fractional polynomials were used to determine the models that best describe the relationships between mosquito bites and age in Mbita (**c**) and Balonghin (**d**) and mosquito bites and body surface area calculated using Dubois equation (**b**, data from Balonghin). The lines in **b**–**d** represent predicted numbers of mosquito bites per individual based on selected models (see 'Methods' section). The green (**b**) and purple (**c**, **d**) areas represent 95% confidence intervals

mosquitoes than younger children (<5 years). In Mbita, although children aged <5 years received relatively fewer bites compared to the other age groups, there was no statistically significant difference: IRR 2.3 (95% CI 0.8–6.2) for children aged 5–15 years vs. younger children (reference group) and IRR 2.1 (95% CI 0.7–5.7) for adults relative to children aged <5 years. In Balonghin, weight and height data were collected and used to estimate body surface area. In this setting, higher body surface area was correlated with age and was associated with more frequent mosquito exposure (Fig. 2b).

In both study sites, a non-negligible percentage of mosquito blood meals (15.8% [153/966] in Balonghin and 12.6% [87/689] in Mbita) had multiple human DNA sources, indicative of mosquitoes feeding on two or more individuals the night before collection. Since it was not possible to unequivocally determine the sources of human alleles in these blood meals, mosquitoes with multiple human blood sources were not included in the statistical analyses. We assessed whether these mosquitoes with unknown multiple blood meal sources may have affected our estimations of age-specific biting patterns. For this, we identified alleles ($N = 125$) that were unique to single individuals at the household level based on data from Balonghin, Burkina Faso. Fourteen of these unique alleles were in children aged <5 years living in households where at least one multiply fed mosquito was collected. None of these unique alleles in young children were present in multiple-source blood meals. To further assess the sensitivity of our outcomes to biting preferences of mosquitoes with multiple human blood sources, we re-analysed our data assuming (1) that multiple-source meals were single blood meals equally allocated to study participants living in the household where they were collected or (2) that individuals who had the lowest genetic distances to the genotypes present in these meals were blood sources: under these different assumptions, adults

were still seven to eleven times more likely to be bitten by mosquitoes than young children in Balonghin (Supplementary Note 1).

**Infectivity to mosquitoes by setting, season and age**. Overall, 1209 mosquito feeding assays were performed to assess infectivity of study participants; 39/1209 (3.2%) individuals infected at least one mosquito (Table 2). No mosquitoes (0/3046) became infected in feeding experiments performed in Kilifi during the dry season. In the other surveys, the percentages of mosquitoes that became infected ranged from 0.05 (4/7716) to 1.6% (121/7749) and was highest in Balonghin. Children aged 5–15 years had similar infectivity compared to younger children (odds ratio (OR) 1.02, 95% CI 0.48–2.20, $P = 0.95$) and were more often infectious to mosquitoes than adults, albeit not statistically significantly (OR 2.07, 95% CI 0.88–4.91, $P = 0.10$; Table 3 and Supplementary Fig. 2). In Burkina Faso, where the number of infectious individuals was highest, nearly all infectious individuals (25/27, 92.6%) had falciparum parasites, asexual or sexual stages, detected by intensive, research quality, microscopy effectively screening 200 fields, which is equivalent to ~4000 white blood cells (Supplementary Fig. 3). In Kilifi and Mbita, 2/3 (66.7%) and 7/9 (77.8%) infectious individuals, respectively, had microscopically detectable falciparum parasites. Six out of nine infectious individuals in Mbita surveys had patent *Plasmodium malariae* infections; in Laye, 1/14 infectious individual only had patent *P. malariae* infection (see Supplementary Note 2 and Supplementary Table 1). Among all 31/39 infectious individuals without evidence of co-infection with non-falciparum malaria parasites, *P. falciparum* gametocytes were detected by microscopy in 61.3% (19/31) and by *Pfs25* mRNA QT-NASBA in 100.0% (28/28). Gametocyte densities by *Pfs25* mRNA QT-NASBA were positively associated

**Table 2 Membrane feeding assays**

| Study site | Season | Number of participants | Number of mosquitoes dissected | Number of infectious individuals | Number of infected mosquitoes | Number of mosquitoes dissected per participant[a] | Proportion of infected mosquitoes per participant[b] | Median oocyst count (range) |
|---|---|---|---|---|---|---|---|---|
| Burkina Faso | Dry | 198 | 17,231 | 14 | 110 | 91 (79–97) | 0.04 (0.01–0.27) | 2 (1–22) |
|  | Wet | 196 | 7749 | 13 | 121 | 41 (35–46) | 0.23 (0.02–0.50) | 3 (1–71) |
| Mbita, Kenya | Dry | 202 | 7071 | 7 | 28 | 30 (28–49) | 0.02 (0.02–0.57) | 4 (1–197) |
|  | Wet | 200 | 6842 | 2 | 5 | 30 (27–46) | 0.07, 0.8 | 2 (1–4) |
| Kilifi, Kenya | Dry | 139 | 3046 | 0 | 0 | 17 (11–31) | – | – |
|  | Wet | 274 | 7716 | 3 | 4 | 28 (19–35) | 0.04 (0.03–0.07) | 2 (1–4) |

[a]Median (interquartile range)
[b]Median (range), only data from infectious individuals included

**Table 3 Age-specific contributions to local malaria transmission**

|  | Prevalence of infectiousness % (number of infectious individuals/total number of participants) | | | % of infected mosquitoes (number of infected mosquitoes/number of dissected mosquitoes) | | | Contribution to the pool of infected mosquitoes before adjustment for mosquito exposure (%) | | | Contribution to the pool of infected mosquitoes after adjustment for mosquito exposure (%) | | |
|---|---|---|---|---|---|---|---|---|---|---|---|---|
| Age (in years) | <5 | 5–15 | >15 | <5 | 5–15 | >15 | <5 | 5–15 | >15 | <5 | 5–15 | >15 |
| Burkina Faso |  |  |  |  |  |  |  |  |  |  |  |  |
| Dry | 8.3 (4/48) | 9.0 (9/100) | 2.0 (1/50) | 0.7 (29/4263) | 0.8 (74/8707) | 0.2 (7/4261) | 25.8 | 52.5 | 21.7 | 2.9 | 45.8 | 51.3 |
| Wet | 6.7 (3/45) | 7.0 (7/100) | 6.0 (3/50) | 2.2 (40/1832) | 1.8 (73/3979) | 0.4 (8/1893) | 32.9 | 45.0 | 22.1 | 3.9 | 41.1 | 55.0 |
| Mbita |  |  |  |  |  |  |  |  |  |  |  |  |
| Dry | 4.9 (2/41) | 5.1 (3/59) | 2.0 (2/102) | 0.1 (2/1468) | 0.9 (21/2292) | 0.1 (5/3311) | 6.3 | 69.3 | 24.4 | 2.9 | 73.5 | 23.6 |
| Wet | 0 (0/49) | 2.0 (1/50) | 1.0 (1/101) | 0 (0/1937) | 0.2 (3/1674) | 0.06 (2/3231) | 0 | 57.5 | 42.5 | 0 | 59.7 | 40.3 |
| Kilifi |  |  |  |  |  |  |  |  |  |  |  |  |
| Wet | 3.0 (2/66) | 0 (0/76) | 0.8 (1/132) | 0.2 (3/1850) | 0 (0/2024) | 0.03 (1/3842) | 64.1 | 0 | 35.9 | 46.0 | 0 | 54.0 |

For the high (Burkina Faso) and moderate (Mbita, Kenya) transmission settings, local age-specific exposure to mosquitoes was quantified and used to estimate the contribution of different age groups to the pool of infected mosquitoes. For Kilifi estimates, age-specific mosquito exposure determined in Mbita was used. Estimates in this table include *P. falciparum* mono-infections and mixed or non-falciparum infections

with parasite densities by *18S* qPCR (Fig. 3a; Spearman's rank correlation coefficient for samples with positive *18S* qPCR result 0.35, $P < 0.001$) and positively associated with mosquito infection rates (Fig. 3b). Mosquito infections were mostly observed when feeding on blood containing estimated gametocyte densities ≥10 gametocytes per µL (25/28 infectious feeds). The infection burden in mosquitoes (oocyst density) was positively associated with the proportion of infected mosquitoes (Fig. 3c), as previously reported for experiments with cultured gametocytes[16, 17] and natural infections with *Plasmodium vivax*[18]. High oocyst counts (10 or more oocysts per midgut) were observed in a subset (29/231) of infected mosquitoes in Burkina Faso while in Kilifi all infected mosquitoes had oocyst counts below 5.

At community level, the proportion of individuals who were able to infect mosquitoes was related to transmission intensity: prevalences of infectiousness in Laye, Balonghin, Mbita dry season, Mbita wet season and Kilifi wet season were 4.9%, 6.4%, 3.3%, 1.1% and 0.9%, respectively, after age standardisation based on estimates for Sub-Saharan Africa[19]. In Laye and Mbita, children younger than 15 years represented the majority of infectious individuals during dry season (76.7% and 65.8%, respectively), whereas during wet season, adults and children contributed similar numbers of infectious individuals to the local infectious reservoir. Estimations that consider age-specific mosquito infection rates in feeding assays and demographic data indicate that children below 15 years of age were responsible for most (~78%) mosquito infections in the high endemicity setting in Burkina Faso (Table 3). The relatively high frequency of exposure to mosquitoes of adults indicates that, in Burkina Faso, adults, when infectious, have on average more opportunities to transmit infection compared to children. To estimate the proportion of all mosquito bites in a community on adults, population age structure and age-specific relative exposure to mosquitoes as estimated by our blood meal analyses were used.

Age-specific probability of infection transmission in a single blood meal, determined in feeding experiments, was then used to calculate the proportion of human-to-mosquito transmission events that starts in infected adults. In Burkina Faso, this adjustment for mosquito exposure resulted in a marked change in age-specific contributions to infected mosquitoes with adults contributing ~50% of all transmission events (Table 3).

**Infectivity to mosquitoes in relation to diagnostics.** In our research settings, expert microscopy alone was sufficiently sensitive to detect infections in most infectious individuals (34/39 for all positive feeding assays and 28/31 when excluding non-falciparum [co-] infections). In routine practice, the lowest parasite densities detectable by routine microscopy and rapid diagnostic tests (RDTs) are estimated as 100 and 200 parasites per µL, respectively. When expressing the contribution to the infectious reservoir incorporating these theoretical limits of detection, our findings suggest that routine microscopy and RDT would detect infections in nearly half of all infectious individuals in the different settings studied (Supplementary Table 2). The proportion of infected mosquitoes is the transmission endpoint that is of more relevance to public health. Before adjusting for mosquito exposure, 19.6–52.1% of *P. falciparum* infected mosquitoes became infected from individuals with *P. falciparum* parasite densities below 100 parasites per µL by microscopy, including individuals who had subpatent parasites detectable by PCR (Fig. 4, top panels). After adjusting for mosquito exposure, these percentages increased to 45.4–77.3% (Fig. 4, bottom panels).

**Discussion**

Quantifying the transmission potential of individuals infected with human pathogens is important to guide control strategies. Here, we report a multi-region assessment of malaria

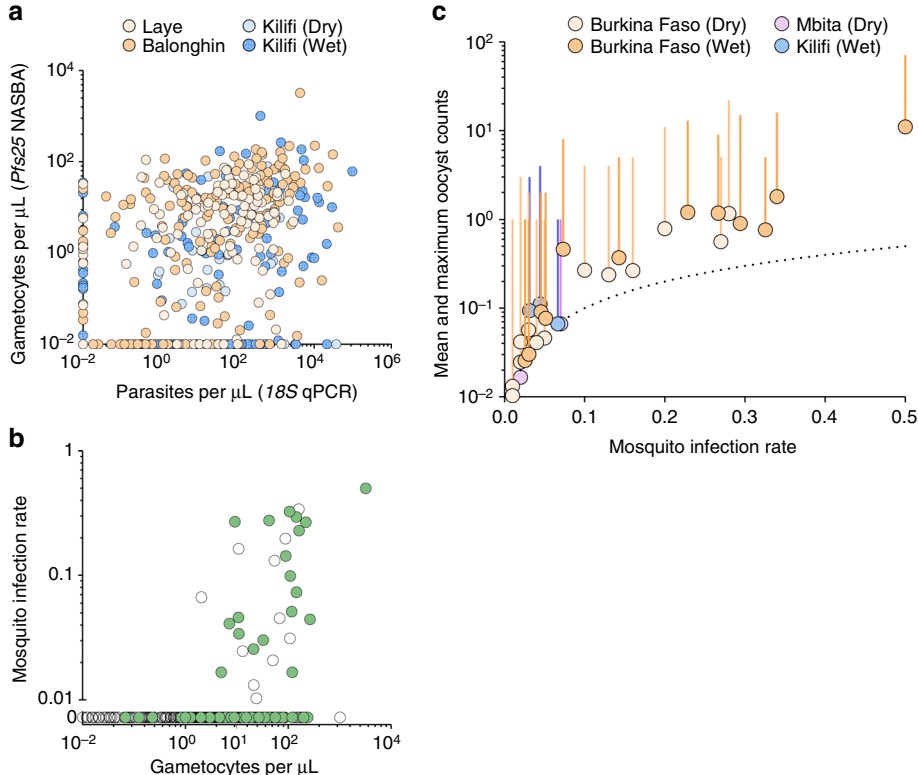

**Fig. 3** Associations between gametocyte and parasite densities, gametocyte density and infectivity, and mosquito infection prevalence and infection burden. In **a**, the association between gametocyte and parasite densities is shown; the y axis is on $\log_{10}$ scale and parasite (*18S* qPCR) and gametocyte (*Pfs25* mRNA quantitative nucleic acid sequence-based amplification (QT-NASBA)) densities were set to 0.01 parasites (or gametocytes) per µL for negative samples to be included in this panel. In **b**, the proportion of mosquitoes infected in individual feeding experiments (y axis) and gametocytes densities (x axis) are shown. Data from all surveys are presented: gametocytes densities were quantified by *Pfs25* mRNA QT-NASBA in samples collected in Burkina Faso and Kilifi, and by microscopy for Mbita participants. Green circles correspond to samples with patent gametocytes. Both the x axis and the segment of the y axis that ranges from 0.01 to 1 are in $\log_{10}$-scale. Individuals who did not infect mosquitoes are presented in a separate segment of y axis that only includes the 0 y coordinate. One infectious individual from Mbita who infected 4/60 mosquitoes had no gametocytes detected by microscopy and no available sample for molecular assays and is therefore not represented in the graph. In **c**, mean and maximum (vertical line) oocyst counts per assay in experiments with at least one mosquito infection are presented. The dotted line represents the hypothetical situation if all infected mosquitoes in an experiment would have exactly one oocyst. In **b**, **c**, feeding experiments where participants had evidence of non-falciparum malaria infections were excluded

transmission using a standardised mosquito feeding protocol and highly sensitive molecular assays for parasite and gametocyte quantification. We observed that only a small proportion of individuals living in malaria endemic areas are infectious to mosquitoes at a given time, and that this proportion decreases with decreasing transmission intensity. While children were more infectious than adults in most surveys, adults were more likely to get bitten by mosquitoes than children and consequently individuals aged >15 years contributed considerably to falciparum infections in mosquitoes.

We performed 1209 mosquito membrane feeding experiments in 1075 individuals of all ages in areas with transmission ranging from intense to low (microscopy-based *P. falciparum* parasite prevalence in participants aged 2–10 years of 34.0%, 51.0%, 40.0% and 15.5% in Laye, Balonghin, Mbita during wet season and Kilifi during wet season, respectively). Overall, we observed that between 1.0 and 7.1% of study participants were capable of infecting mosquitoes at the moment surveys were conducted. This is broadly in line with the limited data available on the human infectious reservoir[14] that are almost exclusively from areas of intense malaria transmission and used both membrane feeding and direct skin feeding assays to measure infectivity. The few studies directly comparing skin feeding vs. membrane

feeding, reviewed in ref. [20], were mostly based on microscopically detectable (high-density) gametocyte carriers and suggest that skin feeding is more sensitive. Higher infection rates in skin feeding assays are nevertheless strongly correlated with infection rates measured by membrane feeding[20]. Skin feeding is biologically attractive since it best reflects natural feeding but is ethically fraught due to discomfort, particularly in children. It is currently unknown what fraction of low-density infections may result in mosquito infections in skin feeding assays but not membrane feeding assays; this information may be of great relevance to translate assessments of transmission by membrane feeding assays to the natural situation. It is noteworthy that our estimates of the proportion of infectious individuals in Burkina Faso in 2013–2014 (4.9 and 6.4%) are lower than those previously estimated in the same setting in 2007–2008[12]. In the previous study, transmission intensity was higher than in the current study with 83–94% parasite prevalence by RNA-based methods and 11–21% of all participants carrying microscopically detectable gametocytes. Lower prevalence and density of parasites will have contributed to the observed difference. Temporal variation in the susceptibility of the mosquito colony to *P. falciparum* may also have contributed. One of the strengths of the current study was that we did not select individuals based on parasite status. Prior

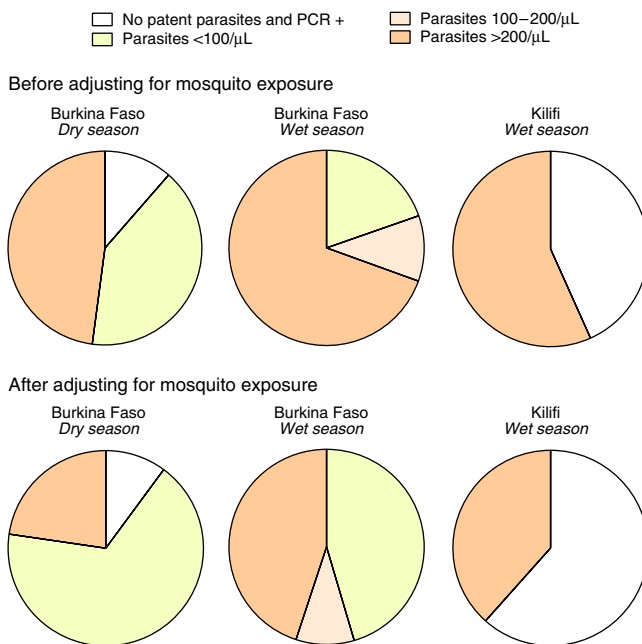

**Fig. 4** Proportion of infected mosquitoes by parasite density. Age-specific prevalences of falciparum malaria parasites by microscopy and PCR and infectiousness prevalences by microscopy-defined parasite density were used to estimate the proportions of *P. falciparum* infected mosquitoes in each community; demographic age structure in Sub-Saharan Africa populations was used to standardise estimates. Individuals with evidence of non-falciparum malaria infections were excluded (*N* = 2). The top panels represent the contributions of human infections with different parasite densities to local mosquito infections, after adjusting for population age structure and age-and-parasite density-specific probabilities of mosquito infection in feeding assays; in the bottom panels, age-specific relative mosquito exposure data were used. These calculations were based on 13, 12 and 3 infectious individuals and 108, 104 and 4 infected mosquitoes in the Burkina Faso dry and wet season surveys and in the Kilifi wet season survey, respectively. Data from Mbita are not presented as most infectious individuals in this setting had *P. malariae* co-infections

screening by molecular assays may have increased the proportion of study participants that was infectious to mosquitoes but would have left uncertainties about the transmission potential of undetected infections[6, 21]. We therefore recruited participants for feeding assays from the general population and successfully used molecular diagnostics in three of four study sites. In our surveys, all infectious individuals with molecular assays results available had parasites detected by *18S* qPCR and *Pfs25* mRNA QT-NASBA, except one infectious individual believed to have transmitted *P. malariae* parasites. This suggests that these assays might be useful to exclude non-infectious individuals. However, it is currently unclear what the kinetics of parasite densities are in chronic submicroscopic infections and conceivable that some infections that are not detectable by these sensitive assays at one time-point may increase in density and likelihood of transmission in the future.

There is accumulating evidence that in all endemicities substantial proportions of falciparum infections are subpatent, i.e. below the limit of detection of conventional field diagnostics[7]. In line with this, we detected a considerably larger number of infections with molecular assays than microscopy. In contrast to findings with high-volume qPCR from a large epidemiological study in Southeast Asia[6], where the percentage of undetectable infections was estimated based on distributions of quantifiable

parasite densities, we found no evidence for a significant number of infections being missed by *18S* qPCR, as indicated by Fig. 1b. There is considerable interest in quantifying the contribution of low density, submicroscopic, infections to onward transmission. Submicroscopic infections are defined as infections that are detectable by molecular methods but not by microscopy. In practice, this definition is influenced by the sensitivity of microscopy and molecular assays, both of which can vary between settings. In our two surveys in Burkina Faso, parasite densities below 100 parasites per μL were detected by research quality microscopy in 35.2 and 41.5% of infectious individuals, who were responsible for 45.4 and 67.2% of infected mosquitoes (Fig. 4), suggesting that a non-negligible proportion of transmission events may be missed by routine microscopy but not necessarily by research microscopy where a larger number of microscopic fields are screened (200–400 fields in our study). Both parasite quantification by microscopy and qPCR have limitations[22, 23] and ultimately the detectability of the infectious reservoir may need to be judged against diagnostic practices that are relevant to guide interventions in the field. If transmissible low-density infections could be targeted by interventions using improved diagnostics, such as highly sensitive RDTs, or that include individuals irrespective of parasite status, transmission might be reduced more effectively and rapidly. Of note, in Kilifi, one individual of three who were infectious in feeding experiments did not carry patent parasites. Whilst we believe the low proportion of infectious individuals accurately reflects the low likelihood of transmission in this setting, numbers are limited to draw conclusions on the performance of different diagnostics to identify the human infectious reservoir for malaria. For this, the methodology for xenodiagnostic studies may need to be refined to include sensitive screening tools to identify potentially infectious individuals in low transmission areas and provide more robust estimates of population infectiousness.

The probability of infecting at least one mosquito and the proportion of infected mosquitoes in successful feeding experiments were positively associated with gametocyte density[21, 24]. Among infectious individuals with *P. falciparum* mono-infection, 61.3% were gametocyte positive by microscopy despite screening 200 microscopic fields specifically for gametocytes, whilst 100.0% were gametocyte positive by *Pfs25* mRNA QT-NASBA. In our surveys, mosquito infection rates were loosely associated with gametocyte densities[21, 24] and most infectious individuals had an estimated density of 10 or more gametocytes per μL by *Pfs25* mRNA QT-NASBA. Since children harbour the highest parasite and gametocyte densities, they are generally considered to constitute a large fraction of the infectious reservoir.

Our observations also confirm that children are more likely to infect mosquitoes than adults. In the high endemicity setting in the current study, the proportion of mosquitoes acquiring malaria infection in feeding experiments was fourfold higher when feeding on blood from children below 15 years of age compared to older individuals (Table 3). This is in broad agreement with previous findings from the same setting (fivefold higher infection rates in children)[12]. In the lower endemicity settings in the current study, mosquito infections were rare and occurred from children and adults without any obvious age dependency; however, in Kilifi the number of mosquitoes dissected per assay was comparatively low, particularly during the dry season survey, which could have prevented the identification of age-related patterns of infectiousness and may have resulted in a lower sensitivity to detect sporadic mosquito infections.

One of the novel elements of our study is that we determined actual *Anopheles* mosquito exposure in areas where xenodiagnostic surveys were performed. Earlier field quantifications of human attractiveness to malaria vectors[25–27] utilised a series of

different techniques, ranging from direct observation to blood meal typing using markers with limited polymorphism (e.g. ABO group). Our study expanded these observations by collecting mosquitoes in households with variable numbers of inhabitants in two different endemic settings. We were able to uniquely link 1094 mosquito blood meals to household occupants. We observed different patterns in mosquito feeding choices in Balonghin (Burkina Faso) vs. Mbita (Kenya) that have consequences for our interpretation of the human infectious reservoir for malaria. In Balonghin, adults were twenty times more likely to be sampled by mosquitoes than children aged <5 years; in Mbita, adults received twice as many bites compared to young children. The difference in mosquito feeding choices between the sites may be partially explained by differences in bed net use; in Mbita, adults reportedly slept under nets more often (87.7%) than children <5 years (77.8%) and schoolchildren (63.8%), while in Burkina Faso, reported bed net usage was similar in all age groups (90.0%, 87.1% and 88.6% for young children, school children and adults, respectively). Differences in species composition of local vector populations is another possible explanation, although evidence for between-*Anopheles* species variation in feeding behaviour with regards to human hosts at the individual level is limited[28]. Although we also observed considerable variation in mosquito exposure between households, and ideally we would have been able to quantify mosquito exposure for every xenodiagnostic survey participant, our observations of mosquito feeding preferences in a selection of households allow a better interpretation of mosquito feeding experiments by extrapolating from standardised numbers of mosquitoes in feeding assays to actual mosquito sampling rates that are strongly age dependent. When adjusting our estimates for mosquito exposure, the relative contribution of adults to the infectious reservoir nearly doubles in Burkina Faso and resulted in a more modest increase in the contribution of older individuals to the infectious reservoir in our Kenya sites. Taken together, our results highlight that even in highly endemic settings where children account for the vast majority of clinical malaria episodes, control interventions that target based on infectiousness may require inclusion of adults.

Two unanticipated findings were the high proportion of *P. malariae* co-infections in Mbita and, to a lesser extent, in Burkina Faso and the high proportion of multiple-human-source blood meals in wild-caught mosquitoes. We only determined non-falciparum malaria co-infections based on microscopy. Screening for non-falciparum infections by molecular methods may have increased the estimates of the parasitaemic reservoir of *P. malariae* and *Plasmodium ovale*[29] (Supplementary Note 2).

Overall, 14.5% of all mosquito blood meals contained DNA from more than one human source, indicative of repeated (potentially interrupted) feeding during the night prior to sampling. This relatively high percentage of bloodfed mosquitoes with multiple human sources of blood is likely to impact comprehensive malaria transmission models[30]. Currently these do not explicitly account for multiple feeding behaviour, despite previous reports[31, 32]: if sporozoites are inoculated with every probing event, mosquito multiple feeding behaviour is likely to increase the risk of human malaria infection. A number of mosquito blood meals (14.4% in Balonghin and 25.2% in Mbita) could not be linked to residents of study houses. In Balonghin, nearly all household occupants provided a blood sample that allowed genetic matching to mosquito blood meals and this suggests indoor resting of mosquitoes that fed elsewhere. In Mbita however, the higher percentage of unmatched mosquitoes could be at least partially explained by the fact that 13.3% (25/188) of household occupants did not provide blood samples for matching. A limitation of our mosquito exposure estimates is that only malaria vectors resting indoors were collected and outdoor biting may account for a non-negligible proportion of mosquito exposure events. One might assume that outdoor mosquito exposure is associated with increasing age, reflecting sleeping patterns, and our finding of a disproportionate number of mosquito bites encountered by adults may thus be an underestimate of true differences in mosquito exposure between age groups.

Individuals recruited into the xenodiagnostic surveys, as with most community-based cross-sectional surveys, were primarily asymptomatic. One study subject in Laye developed acute malaria that required immediate treatment prior to feeding assay and was not sampled, whilst three study participants in Balonghin reported recent (within a week of enrolment) antimalarial treatment, including two individuals, one infectious and one non-infectious, with *Plasmodium ovale* co-infection. In Kilifi, five individuals had positive RDT results (parasite densities by microscopy, range 1240–21,500 parasites per μL) and body temperatures higher than 37.5 °C; one of these individuals infected mosquitoes in feeding assays, the others participated in feeding experiments but were non-infectious. Data on recent or current malaria symptoms were not collected for Mbita participants. Our observation that gametocyte carriage is common in asymptomatic infections, 44.2–90.0% by *Pfs25* mRNA QT-NASBA in those individuals with *18S* qPCR positive results in our surveys, together with evidence from a previous study in Mbita where asymptomatic gametocytaemic children were substantially more infectious than symptomatic gametocytaemic children[33], suggest that individuals without malaria symptoms are likely to contribute more to local transmission compared to symptomatic individuals. Acute symptomatic infections with high parasite densities are likely to be characterised by a shorter duration of infection and thus shorter time-window that allows gametocyte production compared to infections that are more chronic in nature[34]. Despite these considerations that support the importance of asymptomatic infections for the human infectious reservoir, it is of great relevance to directly compare onward transmission from symptomatic and asymptomatic infections in the same setting. Such studies require a design where participants are recruited from both local health facilities and the communities they serve.

In summary, our study that combined actual mosquito exposure assessments and xenodiagnostic surveys indicates that ~45–75% of all mosquito infections are caused by individuals with total parasite densities below 100 parasites per μL but gametocyte densities above 10 gametocytes per μL. These densities are not generally detected by routine diagnostics. Because of their higher exposure to mosquitoes, adults contribute much more to local transmission in some settings than expected based on age-specific infectiousness prevalences and adjustments for population age structure. These findings highlight the potential roles for sensitive diagnostics and interventions that target all age groups to reduce the transmission of malaria across different endemicities.

## Methods

**Study areas and populations**. Our infectivity surveys were conducted in an area of intense malaria transmission in Burkina Faso and areas of low and moderate malaria transmission in Kenya. In Burkina Faso, the dry season survey was performed in the village of Laye and the wet season survey in nearby Balonghin. Both villages are characterised by intense seasonal transmission[12, 35]. In Kenya, dry and wet season surveys were performed in Mbita, on the shores of Lake Victoria in the Suba District in Western Kenya, an area characterised by moderate malaria transmission intensity[36]. In Kilifi, in coastal Kenya, malaria transmission occurs throughout the year, but peaks during the wet season[37]. Over the last 25 years, malaria incidence in this area has declined to very low levels[38]. Of note, health facilities, clinics, were available in all study villages, and artemisinin-based combination therapy is used as first-line treatment of clinical cases in all sites. At the time of these surveys, seasonal malaria chemoprophylaxis for children under 5 years of age was not part of national guidelines in Burkina Faso.

In all xenodiagnostic surveys, children aged 2 years or older and adults were recruited regardless of their parasite status. Individuals with serious clinical conditions requiring immediate treatment were not eligible and were referred to the nearest health facility. Different population sampling strategies were adopted in different surveys. In Laye, recent census data were not available, and the random walk method[39] was used to select households ($N = 59$); each selected household contributed 4 (median, IQR 3–4) study participants, children and adults. In Balonghin (Burkina Faso), a census list was used to randomly select households ($N = 81$); individuals living in these households were randomly selected to participate in the study. For the cross-sectional study performed in Mbita during the dry season, a different sampling strategy was used: the study area was divided in sub-areas with ~10 households; the study team visited a sub-area per day and randomly selected up to six subjects to membrane feeding experiments among those individuals willing to participate in the study. During the following wet season, whenever possible, the same individuals that participated in the first Mbita survey were recruited. In Kilifi, individuals were invited to come to the study clinic and, as with the other study sites, were enroled regardless of their parasite status.

**Sampling strategy**. The objective of these infectivity surveys was to estimate the prevalence of infectiousness in different age groups and transmission settings. Due to logistical constraints related to mosquito husbandry and dissection capacity, we estimated that we would be able to perform on average six membrane feeding assays per day in the different study sites. Two hundred participants were recruited during each survey in Burkina Faso and Mbita; in Kilifi, seasonality is less marked and 413 individuals were recruited over a period of 14 months. Children aged 5 years or younger, schoolchildren (between 5 and 15 years) and adults (15 years or older) were recruited in a 1:2:1 ratio in surveys performed in Burkina Faso; in Kilifi and Mbita, where adults were hypothesised to form a more important part of the parasitaemic reservoir, a ratio of 1:1:2 was targeted.

**Parasite detection**. In our infectivity studies, malaria parasites (asexual and sexual stages) were quantified by light microscopy and molecular assays. All microscopy slides were double-read and considered negative if no parasites were detected in 100 (Burkina Faso and Kilifi) or 200 (Mbita) microscopic fields. Densities of falciparum and non-falciparum malaria parasites were determined by assuming 8000 white blood cells per μL of blood (Supplementary Note 3).

Finger prick blood samples collected in Laye, Balonghin and Mbita were stored in RNAprotect Cell Reagent and had nucleic acids extracted using MagNAPure LC automatic extractor (Total Nucleic Acid Isolation Kit—High Performance, Roche Applied Science). All samples from Kilifi were collected by venipuncture: DNA was extracted from 100 μL of whole blood using the Qiagen extraction method on an automated Qiaxtractor (Qiagen); for RNA based assays, 100 μL of whole blood was stored in the appropriate volume of Trizol (Invitrogen) and extracted using the phenol-chloroform method. 18S rRNA QT-NASBA was used to detect all falciparum parasite stages. Nested PCR targeting the 18S rRNA gene was also performed to detect falciparum infections[40]; and qPCR (18S qPCR) allowed estimation of P. falciparum parasite burden. The detection and quantification of mature P. falciparum gametocytes was performed by Pfs25 mRNA QT-NASBA according to the protocol described in ref. [24].

**Assessment of infectiousness**. Venous blood samples collected in heparinized tubes were used for whole blood membrane feeding assays following an established protocol[41]. Immediately after venipuncture, 400–500 μL of blood were offered to female A. gambiae mosquitoes via an artificial membrane. After 20 min, fully fed mosquitoes were transferred to storage cups by aspiration and kept at 29 °C on average for 1 week (6–8 days) with access to glucose solution prior to dissection and microscopical assessment for the presence of oocysts.

Mosquito infections in Burkina Faso and Kilifi surveys were confirmed by molecular methods. Midguts of mosquitoes with at least one oocyst identified by microscopy were stored in 50 μL of RNAprotect Cell Reagent. MSP2-based nested PCR[42] was used for mosquitoes infected in Burkina Faso feeding experiments. Genetic material (cDNA) of mosquitoes infected in Kilifi were tested with CSP-based PCR. We were not able to reliably confirm mosquito infection for Mbita feeding experiments due to technical issues with sample storage.

**Quantification of mosquito exposure**. Mosquito collections were performed in Balonghin (Burkina Faso) and Mbita (Kenya). In Balonghin, mosquito collections were performed at three different timepoints: end of the 2013 wet season (November–December), and start (June 2014) and peak (September 2014) of the following wet season. Every week indoor mosquito collections were performed in five households. During the first entomological survey (2013), 40 randomly selected households were included; for each household, mosquitoes were collected during seven days or until 30 bloodfed mosquitoes were collected. In the second and third surveys, mosquitoes were collected over 5 days in 20 households with highest mosquito exposure in the first survey and with no changes in the number of inhabitants. Demographic data are not available for five houses selected only for the 2013 survey and where no bloodfed mosquitoes were obtained; individuals living in these households are not included in the analysis presented here. In Mbita, mosquitoes were collected from 40 households during a single time-point from August to December 2015; mosquitoes were collected during seven days or until 30 bloodfed mosquitoes were collected.

Mosquitoes were aspirated from walls and ceilings between 7 and 9 in the morning for a maximum of 15 min per house, transferred to paper cups, transported to the Centre National de Recherche et de Formation sur le Paludisme (CNRFP) in Ouagadougou or the International Centre of Insect Physiology and Ecology (ICIPE) in Mbita and processed immediately upon arrival to minimise DNA degradation. If blood fed, the abdomen of mosquitoes was squeezed onto a Whatman 3MM filter paper for later DNA extraction. To link mosquito blood meals to specific individuals, all permanent inhabitants of households selected for mosquito sampling were asked to provide a single finger prick blood sample into plastic K2EDTA BD Microtainer for collection of 50 μL-blood samples in 250 μL of RNAprotect Cell Reagent and onto a Whatman 3MM filter paper.

Nucleic acids were extracted from filter papers with squeezed bloodfed mosquito abdomens and with human samples from Balonghin study participants using Boom extraction method[43]. Nucleic acids from human blood samples collected in Mbita in RNAprotect Cell Reagent were extracted using MagNAPure LC automatic extractor (Total Nucleic Acid Isolation Kit—High Performance, Roche Applied Science).

Molecular analysis of blood meals was performed using Authentifiler PCR Amplification kit (Applied Biosystems), which consists of a multiplex PCR assay with nine microsatellite markers and one gender-determining marker (Amelogenin). Automated capillary electrophoresis was used to determine DNA profiles in mosquito blood meals. 88/1066 and 72/808 mosquito samples collected in Balonghin and Mbita respectively had no amplification. A small number of samples with amplification had <10 alleles detected (12 in Balonghin and 47 in Mbita) and were not included in our analysis. In total, 153 mosquito samples from Balonghin and 87 from Mbita had evidence of multiple human DNA sources, i.e. more than two alleles in at least three loci (see Results section and Supplementary Note 1, including Supplementary Tables 3 and 4). This conservative definition of multiple feeding was to avoid artefacts due to minor peaks in capillary electrophoresis resulting from variable amounts of human DNA. Of those mosquito blood meal samples that had a single human DNA source, 139 and 174 in Balonghin and Mbita respectively were not matched to someone living in the same household where they were collected and for this reason were also excluded from our analysis; additionally eight mosquitoes collected in Burkina Faso did not have identification data available.

**Statistical analysis**. Statistical analyses were performed in Stata 14 (StataCorp LP, Texas, USA). Prevalences of falciparum parasites and gametocytes were estimated for each xenodiagnostic survey. Age-specific (<5 years, 5–15 years and >15 years) asexual stage parasite, gametocyte and infectivity prevalences were also calculated. We compared parasite levels between children and adults and gametocyte levels during wet season surveys vs. dry season surveys using negative binomial regression. Logistic regression was used to assess the association between age and infectiousness (at least one infected mosquito in feeding experiments). In these models, estimates were adjusted for study site. In Fig. 4, parasites densities in patent infections were defined based on microscopy results; the highest stage-specific density (asexual stage levels or gametocytaemia) in individual infections was used. In Fig. 1, two participants of the Kilifi wet season survey had parasite densities above 100,000 parasites per μL by 18S qPCR, ~10–30 times higher than microscopy-based estimates, and microscopy-defined densities were presented for these individuals. In Fig. 2, fractional polynomials were used to determine the combinations of powers that best describe the associations between age or body surface area and number of mosquito bites; mixed effects negative binomial models with household as random effect were fitted. The lines in Fig. 2b, d correspond to the predicted number of mosquito bites per individual; for Balonghin (b, d), this number represents mosquito exposure during peak wet season over 5 days.

An algorithm was developed that quantified squared differences in allele sizes for each locus between human and mosquito samples and identified study participants with the lowest mean allelic distance to each mosquito blood meal. All blood meal samples from the Burkina Faso study and a subset of mosquito samples from Mbita had their matching result checked manually. The effect of age, as a categorical variable, on mosquito exposure (i.e. the number of singly matched mosquito blood meals to an individual) was assessed by mixed effects negative binomial regression. Initially, we fit a model that included data from both study sites and tested for interaction between age effects and study site effects. Since there was a significant interaction ($P = 0.001$) between study site and age effects, we present separate analyses for the different sites. For the entomological study in Balonghin, this analysis was adjusted for survey (fixed effect). To account for data correlatedness and household-level differences in vector abundance, the models for both sites had household as random effect.

Age-specific contributions to transmission were estimated as (i) the proportion of infectious individuals from each age group and (ii) the proportion of infected mosquitoes acquiring infection from each age group.

The relative contribution ($C_i$) of each age group $i$ to the pool of infective individuals in each study site was calculated:

$$C_i = p_i d_i / \sum_1^n p_i d_i, \tag{1}$$

where $d_i$ is the proportion of the population in age group $i$, $p_i$ is the age-specific prevalence of infectivity (at least one infected mosquito in feeding experiments) and $n$ is the number of categories into which the population was divided. Values of $p_i$ are study site-specific. For this analysis three age categories were used ($n = 3$, <5 years, 5–15 years and >15 years) and we used an age structure based on recent estimates for Sub-Saharan Africa[19].

To estimate the proportion of infected mosquitoes that acquire their parasites from a specific age group ($M_i$) (Table 3), the following formulas were used:

$$b_i = d_i a_i / \sum_1^n d_i a_i \tag{2}$$

$$M_i = w_i b_i / \sum_1^n w_i b_i \tag{3}$$

where $b_i$ is the proportion of all mosquito bites which occur on age group $i$, and $a_i$ represents the age-specific relative exposure to mosquitoes, which was calculated based on the results of mosquito blood meal analysis (for Burkina Faso study sites, $a_1 = 1$, $a_2 = 7.7$, $a_3 = 20.9$; for Mbita surveys, $a_1 = 1$, $a_2 = 2.3$, $a_3 = 2.1$); $w_i$ corresponds to the probability of mosquito infection (number of mosquitoes infected divided by total number of mosquitoes dissected) in feeding assays involving individuals from age group $i$, regardless of their parasite status.

We also determined the contribution to transmission of individuals with different parasite densities ($M_j$) (Fig. 4). Equation 3 was modified by re-assigning mosquito bites by age and diagnostic status using age-specific prevalences of infections with different parasite levels ($t_{ij}$), where $i$ corresponds to specific age groups and $j$ to parasite density categories:

$$M_j = \sum_{i=1}^n w_{ij} b_{ij} t_{ij} / \sum_{j=1}^m \sum_{i=1}^n w_{ij} b_{ij} t_{ij} \tag{4}$$

Here, $m$ represents the number of different diagnostics categories: $j = 1$ (no patent parasites but PCR positive), $j = 2$ (patent infections with <100 parasites per μL), $j = 3$ (patent infections with 100–200 parasites per μL) or $j = 4$ (patent infections with >200 parasites per μL). In this modified framework, $w_{ij}$ represents the age ($i$) and parasite density category ($j$) specific probability of mosquito infection in feeding assays.

**Ethics**. The study received ethical clearance from local and international ethical committees: xenodiagnostic and entomological studies in Burkina Faso were approved by the ethical review committees of the Ministry of Health of Burkina Faso and of the London School of Hygiene and Tropical Medicine (reference numbers 6271 and 6447); xenodiagnostic surveys in Mbita and Kilifi, Kenya, were combined in one protocol and received clearance from the Kenya Medical Research Scientific Ethics Research Unit (KEMRI-SERU SSC number 2574). Written informed consent was obtained from all participants or the parents/guardians. The study was conducted in accordance with the principles of the Helsinki Declaration.

**Data availability**. The infectivity dataset generated and analysed during the current study is available in the Dryad repository (doi:10.5061/dryad.c3n63); all other datasets are available from the corresponding author on reasonable request.

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

## Acknowledgements

This work was supported by the Bill and Melinda Gates Foundation (AFIRM OPP1034789). T.B. is further supported by a fellowship of the Netherlands Organization for Scientific Research (VIDI fellowship grant number 016.158.306). We would like to thank the clinical and entomology teams at CNRFP (Burkina Faso), St Judes Clinic (Mbita) and KEMRI-Wellcome Trust (Kilifi), and the participants of the different studies and their families.

## Author contributions

C.D. and T.B. designed the study. B.P.G., M.C.K., C.D. and T.B. wrote the first draft of the manuscript. B.P.G., M.C.K., J.B. and T.B. analysed the data. B.P.G., W.M.G., A.B.T., I.N., S.B.S. and T.B. contribute to data collection in Burkina Faso. M.C.K., J.M.N., D.M., J.M., K.M. and P.B. contributed to data collection in Kilifi. P.S., W.S., G.J.H.B., R.O. and M.N. contributed to data collection in Mbita. K.L., J.M.N. and D.M. performed molecular assays for parasite detection and quantification. L.G. performed mosquito blood meal genotyping; J.H. developed algorithm used to match blood meals to humans. C.D. and T.B. led the study team. All authors contributed to interpretation of the analyses and revised the draft manuscript.

## Additional information

**Competing interests:** The authors declare no competing financial interests.

