## [Peer Review File · Nature Communications]

Reviewers' comments:

Reviewer #1 (Remarks to the Author):

Bousema and colleagues combine molecular parasite detection methods with population-level mosquito feeding surveys to understand what proportion of the asymptomatic human infectious reservoir is captured by current malaria field diagnostics (microscopy, RDT). They also perform mosquito captures and bloodmeal genotyping to measure age-specific biting preferences, a novel strategy that adds another layer to their findings. Their findings, that infectivity correlates with transmission intensity, and that greater mosquito sampling of adults counterbalances the greater infectiousness of children vs. adults, are relevant to the design of strategies to interrupt malaria transmission. They should be of interest to a wide audience.

Their methods are sound, data is well presented, and conclusions for the most part justified. I think 2 major limitations of the study that are not cited are the low infectivity rate in the low transmission setting (so not enough sample size to draw conclusions) and the use of membrane feeding assays to measure infectiousness of natural infections.

Major comments:

1. Relatively low infectivity rates, especially in low transmission setting. Conclusions are limited by rates of mosquito infectivity that were perhaps less than expected, especially in low-transmission Kilifi. This is glossed over in the abstract by saying that less than 10% of individuals at all study sites were infectious. It would be clearer to say that out of the 1209 feeding assays performed, 39 or 3.2% were infectious. Or that 6.9% (27/394) vs. 1.7% (3/ 413) of individuals in the high vs. lowest transmission settings were infectious. While the mosquito infection rates are given, it is more telling and easier to understand the differences in person terms. You could also use the age-adjusted rates in line 248.

Similarly, I initially took "1/3" as one-third. It would be clearer if stated as "1 of 3... was detected by molecular techniques only." Or use the same construction as the prior sentence, "whilst 1 of 3 was not detected by microscopy." As currently stated, a lot of emphasis is placed on missing 1 person who infected 1 mosquito if molecular diagnostics are not used in the low transmission setting (which has the most relevance to areas pursuing elimination). This limitation of small numbers of transmitters in the low transmission setting should be discussed.

The differences from their previous study in Burkina Faso (J Infect Dis 2015) which reported 33% of feeds were infectious (vs 7% here), and infected 7.6% of mosquitoes overall (vs 1.6% here) are not mentioned in the discussion.

2. Measuring infectiousness in the low transmission setting. With such a low population-level rate of infectiousness, it may become relevant that direct skin feeding assays were not used to identify infectious persons. Also in Table 2, it seems that in over half of the individuals in Kilifi, <30 blood-fed mosquitoes were dissected, perhaps biasing to an underestimate of the number of infectious individuals. This limitation should be noted.

3. Lines 434-442: Asymptomatic vs symptomatic transmission.

Whilst assessments of the relative contribution of clinical
435 malaria cases and asymptotically infected individuals to local transmission are relevant
436 13,18, the vast majority of parasite infections identified by our cross-sectional surveys was
437 without symptoms prompting healthcare seeking behaviour. Together with evidence from a
438 previous study in Mbita where asymptomatic gametocytaemic children were substantially
439 more infectious than symptomatic gametocytaemic children 30, these observations indicate
440 that the contribution of individuals developing malaria symptoms to transmission might be
441 largely overwhelmed by the contribution of those infected without symptoms, especially
442 during dry seasons.

I find this discussion point speculative. The one study that is cited had high rates of infectivity in both asymptomatic and symptomatic gametocyte carriers not seen in this study. Also, the present

study was not designed to capture symptomatic patients. In Kilifi, it is unlikely that symptomatic persons would present to the study clinic to join the study (lines 477-8). Not taking into account the relative sizes of the symptomatic and asymptomatic reservoirs, it is hard to believe that symptomatic gametocyte carriers would infect fewer mosquitoes than the 3 individuals in Kilifi who infected 4 mosquitoes (how many had microscopic gametocytes?), since symptomatic carriers with higher parasite densities are more likely to have microscopic gametocyte densities. Furthermore, in the present study no infectious individuals were found in Kilifi during the dry season.

Minor comments:

1. It might be worth noting in the discussion that 18S qPCR detected all infectious individuals, arguing against the need for ultrasensitive assays outside of research studies.
2. Line 252 – sentence not finished.
3. Lines 360-363 – sentence fragment.

Reviewer #2 (Remarks to the Author):

This is a very well-written paper on a novel analysis that is of great relevance to the field. I strongly support its publication.

Below please find suggestions that I think will increase the impact of the paper on its audience.

- L87 The wording should be modified here. "Conflicting indications" suggests a lack of reproducibility between previous experimental results. But the two previous citations have very different sampling frames -- cross-sectional (Burkina) vs. uncomplicated malaria incidents (Cambodia) -- which results in different distributions of gametocyte density and possibly also different relationships between density and infectiousness.

- L89 Related to the above comment, the "need for more assessments of infectivity" should consider not just "a range of endemicities" but also a sufficient quantification of parasite and gametocyte densities (and other factors) to inform a coherent understanding of the mechanisms responsible for variation in human-to-mosquito infectiousness.

- L105 What months do the dry and wet season samplings correspond to? This doesn't appear in Methods either (L457)? Although it is stated for the mosquito sampling (L533).

- L126 Supplemental Figure 3 is very useful and might be cited already in the discussion of Figure 1b.

- L133 It is hard to see how the 139 entries in the Kilifi (dry) distribution (Fig 1b), especially the lowest few bins with ~10 entries each can abruptly drop to 0 without a different threshold being applied to that data. Did I miss something in the Methods (L501 vs. L504)?

- L116 Some statement on the expected case-management rates in the different sites and their differential impact on parasite prevalence at different seasons depending on endemicity (Fig 1a) would be appreciated.

- L174/L389 The discussion of age-specific ITN usage is appreciated, but can you also comment on any differences in mosquito species collected between Mbita and Balonghin? Are there any differences in household composition? Is it not possible to present these results (Fig 2) in categories of, e.g. the largest household member, another adult, the largest child, etc.? It is not clear how the dominant feature of Fig 2b, household-to-household variability, interacts with mosquito feeding choices and the observed age patterns.

- L201/L567 In the several hundred mixed bloodmeals, does the distribution of number of loci with

multiple alleles (e.g. 3/10 vs. 10/10) give some indication on the likelihood that the multiple feeds were on members of the same nuclear family within the same household? Regardless, this distribution would be an interesting addition to the supplement.

- L243 More information on the very high oocyst-count infection would be useful (Fig 3C), e.g. what is the gametocyte density measured by microscopy. Also the 197-oocyst infection in Mbita. Perhaps two more columns in Table S1 (microscopy asexual and gametocyte densities)?

- Fig 3b would be more informative if it weren't so squished. Might I recommend coloring by microscopic gametocytemia (the current coloring is redundant); stretching horizontally; and changing to the y-axis to a log-scale down to the lowest non-zero value, then an axis break, then a zero bin.

- L290 Given the large fraction of infections arising from 10-100/uL microscopic density infections, a mention of high-sensitivity RDTs might be relevant here.

- Fig 4 The title of Figure 4 should probably be something more like "Proportion of infected mosquitoes by parasite density". The reader should not have to jump to Table 2 to get a sense of the significance of these results. Adding $N_{\text{humans}}=14$, $N_{\text{mosquitoes}}=110$ directly on the figure for Burkina Faso dry season, etc. would be an improvement.

- L296 (and elsewhere) Please be careful to state clearly what densities (asexual, gametocyte, microscopy, NASBA) are being used in different places. In this example, the specifics are in Methods L584-586 but that leaves the reader guessing what is being shown in the results and Figure 4.

- L324 "multi-site" here is being used to contrast with previous work done in multiple sites separated by a few 10s of kilometers in Burkina Faso [Ref 12]. Multi-country or -region might be more clear?

- L339 The justification of feeding without screening is acceptable here. But before L343 ("There is accumulating evidence...") one has to finish this line of reasoning and make clear that 0/30 infectious humans were 18S- and 25S-, although in Mbita we can't say. Then move on to a new paragraph related to subpatent infections.

- L347 To strengthen this point, it might be worth mentioning that the SE Asian findings involved both fitting the low end of the distribution and imputing the Pf/Pv allocation of unspiciated low density samples based on the spiciated ratio.

- L355 This sentence is a bit of a non sequitur. If you feel it's an important point, it needs to be followed with a reference to Fig S3 and a discussion of different contributions to measurement uncertainty (variable white blood cell counts, log-normal errors from amplification, etc.)

- L357 (related to earlier comment) If you keep the 100/uL qPCR density sentence that precedes this, it should be reinforced that the next sentence is referring to 100/uL by microscopy (greater of asexual and gametocytes as per Methods).

- L370 Greater than 10/uL by what detection method?

- L570 Are the houses where individuals declined to participate in the human blood typing (15% in Mbita) included in the numbers of unmatched bloodfed mosquitoes?

There are a few punctuation issues and awkward sentences in the text. A few examples below:

- L64 "infectious diseases, and ..."

- L73 "The premise being..." (fragment)
- L338 "from areas" "in areas" ??

Reviewer #1 (Remarks to the Author):

Reviewer's comment

Bousema and colleagues combine molecular parasite detection methods with population-level mosquito feeding surveys to understand what proportion of the asymptomatic human infectious reservoir is captured by current malaria field diagnostics (microscopy, RDT). They also perform mosquito captures and bloodmeal genotyping to measure age-specific biting preferences, a novel strategy that adds another layer to their findings. Their findings, that infectivity correlates with transmission intensity, and that greater mosquito sampling of adults counterbalances the greater infectiousness of children vs. adults, are relevant to the design of strategies to interrupt malaria transmission. They should be of interest to a wide audience.

Their methods are sound, data is well presented, and conclusions for the most part justified. I think 2 major limitations of the study that are not cited are the low infectivity rate in the low transmission setting (so not enough sample size to draw conclusions) and the use of membrane feeding assays to measure infectiousness of natural infections.

Major comments:

1. Relatively low infectivity rates, especially in low transmission setting. Conclusions are limited by rates of mosquito infectivity that were perhaps less than expected, especially in low-transmission Kilifi. This is glossed over in the abstract by saying that less than 10% of individuals at all study sites were infectious. It would be clearer to say that out of the 1209 feeding assays performed, 39 or 3.2% were infectious. Or that 6.9% (27/394)

vs.1.7% (3/ 413) of individuals in the high vs. lowest transmission settings were infectious. While the mosquito infection rates are given, it is more telling and easier to understand the differences in person terms. You could also use the age-adjusted rates in line 248.

Similarly, I initially took “1/3” as one-third. It would be clearer if stated as “1 of 3... was detected by molecular techniques only.” Or use the same construction as the prior sentence, “whilst 1 of 3 was not detected by microscopy.” As currently stated, a lot of emphasis is placed on missing 1 person who infected 1 mosquito if molecular diagnostics are not used in the low transmission setting (which has the most relevance to areas pursuing elimination). This limitation of small numbers of transmitters in the low transmission setting should be discussed.

The differences from their previous study in Burkina Faso (J Infect Dis 2015) which reported 33% of feeds were infectious (vs 7% here), and infected 7.6% of mosquitoes overall (vs 1.6% here) are not mentioned in the discussion.

Answer to reviewer’s comment

We characterized malaria transmission potential across high and low endemic African sites and observed very low transmission potential in the low endemic site in coastal Kenya. Whilst we believe our findings capture the (small) infectious reservoir in our low endemic site, we agree that this warrants caution for some of our conclusions and further agree that our small number of infectious individuals has to be clear throughout the manuscript. We have revised our abstract to use the age-adjusted rates of infectiousness, as suggested by the reviewer, and clearly indicate the numbers that conclusions are drawn from. In addition, we have expanded the *Discussion* section to highlight statistical limitations of the study that quantified infectiousness in a low transmission setting, where very low mosquito infection rates are a reality.

We have thus modified the *Abstract* to mention age-standardised prevalences of infectiousness in the different settings and to clarify the “1/3” statement (see comment above).

“Detailed understanding of the human infectious reservoir is essential to rationally target malaria transmission-reducing interventions. We report on the first multi-region study to

assess population-wide malaria transmission potential based on 1,209 mosquito feeding assays in endemic areas in Burkina Faso and Kenya. Overall, 39 individuals were infectious; age-standardised prevalences of infectiousness in the study sites ranged from 0.9 to 6.4%, and the percentage of infected mosquitoes ~~in the surveys~~ ranged from 0.05% (4/7,716) to 1.6% (121/7,749) and correlated positively with transmission intensity.”

“In the high endemicity setting, most infectious individuals were identified by research-grade microscopy (92.6%; 25/27), whilst 1 of 3 infectious individuals in the lowest endemicity setting was detected by molecular techniques alone.”

We have also modified the following paragraph in the *Discussion* section to address the comments above and the similar comments from Reviewer #2. Changes related to the comments above are underlined.

“There is accumulating evidence that in all endemicities substantial proportions of falciparum infections are subpatent, i.e. below the limit of detection of conventional field diagnostics⁷. In line with this, we detected a considerably larger number of infections with molecular assays than microscopy. In contrast to findings with high-volume qPCR from a large epidemiological study in Southeast Asia⁶, where the percentage of undetectable infections was estimated based on distributions of quantifiable parasite densities, we found no evidence for a significant number of infections being missed by 18S qPCR, as indicated by Figure 1b. There is considerable interest in quantifying the contribution of low density, submicroscopic, infections to onward transmission. In practice, this definition is influenced by the sensitivity of microscopy and molecular assays, both of which can vary between settings. In our two surveys in Burkina Faso, parasite densities below 100 parasites per μL were detected by research quality microscopy in 35.2 and 41.5% of infectious individuals, who were responsible for 45.4 and 67.2% of infected mosquitoes (Figure 4), suggesting that a non-negligible proportion of transmission events may be missed by routine microscopy but not necessarily by research microscopy where a larger number of microscopic fields are screened (200-400 fields in our study). Both parasite quantification by microscopy and qPCR have limitations^{22, 23} and ultimately the detectability of the infectious reservoir may need to be

judged against diagnostic practices that are relevant to guide interventions in the field. If transmissible low-density infections could be targeted by interventions using improved diagnostics, such as highly sensitive RDTs, or that include individuals irrespective of parasite status, transmission might be reduced more effectively and rapidly. Of note, in Kilifi, one individual of three who were infectious in feeding experiments did not carry patent parasites. Whilst we believe the low proportion of infectious individuals accurately reflects the low likelihood of transmission in this setting, numbers are limited to draw conclusions on the performance of different diagnostics to identify the human infectious reservoir for malaria. For this, the methodology for xenodiagnostic studies may need to be refined to include sensitive screening tools to identify potentially infectious individuals in low transmission areas and provide more robust estimates of population infectiousness.”

Additionally, we modified the *Discussion* section to mention the results of a previous xenodiagnostic study performed in Burkina Faso, as suggested by the reviewer:

*“We performed 1,209 mosquito membrane feeding experiments in 1,075 individuals of all ages in areas with transmission ranging from intense to low (microscopy-based *P. falciparum* parasite prevalence in participants aged 2 – 10 years of 34.0, 51.0, 40.0 and 15.5% in Laye, Balonghin, Mbita during wet season, and Kilifi during wet season, respectively). Overall, we observed that between 1.0 and 7.1% of study participants were capable of infecting mosquitoes at the moment surveys were conducted. This is broadly in line with the limited data available on the human infectious reservoir¹⁴ that are almost exclusively from areas of intense malaria transmission and used both membrane feeding and direct skin feeding assays to measure infectivity. The few studies directly comparing skin feeding versus membrane feeding, reviewed in²⁰, were mostly based on microscopically detectable (high-density) gametocyte carriers and suggest that skin feeding is more sensitive. Higher infection rates in skin feeding assays are nevertheless strongly correlated with infection rates measured by membrane feeding²⁰. Skin feeding is biologically attractive since it best reflects natural feeding but is ethically fraught due to discomfort, particularly in children. It is currently unknown what fraction of low-density infections may result in mosquito infections in skin feeding assays but not membrane feeding assays; this information may be of great relevance to translate assessments of transmission by membrane feeding*”

*assays to the natural situation. It was noteworthy that our estimates of the proportion of infectious individuals in Burkina Faso in 2013-2014 (4.9 and 6.4%) are lower than those previously estimated in the same setting in 2007-2008. In the previous study, transmission intensity was higher than in the current study with 83 – 94% parasite prevalence by RNA-based methods and 11 – 21% of all participants carrying microscopically-detectable gametocytes. Lower prevalence and density of parasites will have contributed to the observed difference. Temporal variation in the susceptibility of the mosquito colony to *P. falciparum* may also have contributed.”*

Reviewer’s comment

2. Measuring infectiousness in the low transmission setting. With such a low population-level rate of infectiousness, it may become relevant that direct skin feeding assays were not used to identify infectious persons. Also in Table 2, it seems that in over half of the individuals in Kilifi, <30 blood-fed mosquitoes were dissected, perhaps biasing to an underestimate of the number of infectious individuals. This limitation should be noted.

Answer to reviewer’s comment

We have previously reviewed studies that performed matched comparisons of membrane and skin feeding (Bousema et al. *Mosquito feeding assays to determine the infectiousness of naturally infected Plasmodium falciparum gametocyte carriers*. PLOS One 2012). The data are sparse, almost exclusively based on microscopically detected gametocyte carriers and whilst overall data found a clear tendency for skin feeding assays to result in higher mosquito infection rates, it is currently unclear how this may affect the interpretation of infectiousness of low-density parasite carriers. Moreover, whilst our current estimates of infectiousness are lower than those previously reported from Burkina Faso (as the reviewer highlighted in his first comments), they are in line with several other studies. For example, in two xenodiagnostic studies where skin feeding assays were used to assess infectivity (Githeko et al. *The reservoir of Plasmodium falciparum malaria in a holoendemic area of western*

Kenya. *Trans R Soc Trop Med Hyg.* 1992; Muirhead-Thomson. *The malarial infectivity of an African village population to mosquitoes (Anopheles gambiae); a random xenodiagnostic survey.* *Am J Trop Med Hyg.* 1957), ~10% of the population was infectious to mosquitoes, despite parasite prevalence by microscopy higher than 75%. These estimates are similar to ours: in the setting with highest transmission intensity, we observed that 6.4% of the study population was infectious to mosquitoes with an overall parasite prevalence of 52.2% by microscopy. We have, however, modified the *Discussion* section to mention the use of membrane feeding assays and the low number of dissected mosquitoes in Kilifi as possible study limitations:

“We performed 1,209 mosquito membrane feeding experiments in 1,075 individuals of all ages in areas with transmission ranging from intense to low (microscopy-based P. falciparum parasite prevalence in participants aged 2 - 10 years of 34.0, 51.0, 40.0 and 15.5% in Laye, Balonghin, Mbita during wet season, and Kilifi during wet season, respectively). Overall, we observed that between 1.0 and 7.1% of study participants were capable of infecting mosquitoes at the moment surveys were conducted. This is broadly in line with the limited data available on the human infectious reservoir¹⁴ that are almost exclusively from areas of intense malaria transmission and used both membrane feeding and direct skin feeding assays to measure infectivity. The few studies directly comparing skin feeding versus membrane feeding, reviewed in ²⁰, were mostly based on microscopically detectable (high-density) gametocyte carriers and suggest that skin feeding is more sensitive. Higher infection rates in skin feeding assays are nevertheless strongly correlated with infection rates measured by membrane feeding²⁰. Skin feeding is biologically attractive since it best reflects natural feeding but is ethically fraught due to discomfort, particularly in children. It is currently unknown what fraction of low-density infections may result in mosquito infections in skin feeding assays but not membrane feeding assays; this information may be of great relevance to translate assessments of transmission by membrane feeding assays to the natural situation.”

“Our observations also confirm that children are more likely to infect mosquitoes than adults. In the high endemicity setting in the current study, the proportion of mosquitoes acquiring malaria infection in feeding experiments was four-fold higher when feeding on

blood from children below 15 years of age compared to older individuals (Table 3). This is in broad agreement with previous findings from the same setting (five-fold higher infection rates in children)¹². In the lower endemicity settings in the current study, mosquito infections were rare and occurred from children and adults without any obvious age-dependency; however, in Kilifi the number of mosquitoes dissected per assay was comparatively low, particularly during the dry season survey, which could have prevented the identification of age-related patterns of infectiousness and may have resulted in a lower sensitivity to detect sporadic mosquito infections.”

Reviewer’s comment

3. Lines 434-442: Asymptomatic vs symptomatic transmission.

Whilst assessments of the relative contribution of clinical malaria cases and asymptotically infected individuals to local transmission are relevant the vast majority of parasite infections identified by our cross-sectional surveys were without symptoms prompting healthcare seeking behaviour. Together with evidence from a previous study in Mbita where asymptomatic gametocytaemic children were substantially more infectious than symptomatic gametocytaemic children 30, these observations indicate that the contribution of individuals with malaria symptoms to transmission might be largely overwhelmed by the contribution of those infected without symptoms, especially during dry seasons.

I find this discussion point speculative. The one study that is cited had high rates of infectivity in both asymptomatic and symptomatic gametocyte carriers not seen in this study. Also, the present study was not designed to capture symptomatic patients. In Kilifi, it is unlikely that symptomatic persons would present to the study clinic to join the study (lines 477-8). Not taking into account the relative sizes of the symptomatic and asymptomatic reservoirs, it is hard to believe that symptomatic gametocyte carriers would infect fewer mosquitoes than the 3 individuals in Kilifi who infected 4 mosquitoes (how many had microscopic gametocytes?), since symptomatic carriers with higher parasite densities are more likely to have microscopic gametocyte densities.

Furthermore, in the present study no infectious individuals were found in Kilifi during the dry season.

Answer to reviewer's comment

We appreciate the comments related to the unquantified transmission potential of clinical malaria cases and have highlighted this as an area of research that should be prioritized. Current evidence from the study settings and other African settings support our conclusions on the comparative importance of asymptomatic infections but we agree that in particular in low endemic settings, more data are needed on the infectiousness of malaria-infected individuals who present at the clinic with clinical malaria.

Individuals with clinical malaria have on average higher parasite densities compared to individuals who are asymptotically infected though this is not always manifested as higher gametocytes densities due to the maturation period of falciparum gametocytes and timing of treatment. A recent meta-analysis of individual patient data conducted by the WWARN group indicated a strong negative association between asexual parasite density, and fever, in acute malaria cases and gametocyte prevalence, suggesting that acute infections often present without gametocytes while infections that are more chronic and asymptomatic in nature may be associated with a longer production of gametocytes. Moreover, there is evidence from malariotherapy data that gametocytes that are detected in acute infections require several days of maturation to become infectious, further suggesting that, depending on the duration of infection at the moment of clinical presentation, acute infections may not (yet) be as infectious as chronic infections.

In line with this, there are several studies showing the prevalence and density of gametocytes in individuals with clinical symptoms is not necessarily higher than in healthy subjects (another example is Gouagna et al. *Plasmodium falciparum malaria disease manifestations in humans and transmission to Anopheles gambiae: a field study in Western Kenya*. Parasitology, 2004) suggesting that high gametocyte levels might not be frequent in clinical malaria infections that prompt treatment, and that the presence of gametocytes at clinic presentation might be context dependent. In an unrelated study from an area of low endemicity in Ethiopia, we find further evidence that *P. falciparum* gametocyte densities and infectivity to mosquitoes are higher in asymptomatic parasite carriers compared to symptomatic parasite carriers (Tadesse *et al.*, The relative contribution of symptomatic and

asymptomatic *Plasmodium falciparum* and *Plasmodium vivax* infections to the infectious reservoir in Ethiopia, *in preparation*).

Despite these considerations, we would agree that a major component in understanding contributions to the reservoir is the relative numbers of individuals with clinical malaria and with asymptomatic malaria in a local population. It is widely acknowledged from MIS and other community surveys that there are many more asymptomatic infections than symptomatic – particularly with the advent of molecular screening for infection. However, there are only very few studies that simultaneously recruited participants from clinics (symptomatic individuals) and from the community (asymptomatic and symptomatic individuals) to estimate population-wide infectiousness (e.g., Pethleart et al. *Infectious reservoir of Plasmodium infection in Mae Hong Son Province, north-west Thailand*. Malar J. 2004). We have now modified this paragraph of the *Discussion* section to suggest this approach as an important next step towards understanding the infectious reservoir of malaria:

“Individuals recruited into the xenodiagnostic surveys, as with most community-based cross-sectional surveys, were primarily asymptomatic. One study subject in Laye developed acute malaria that required immediate treatment prior to feeding assay and was not sampled, whilst three study participants in Balonghin reported recent (within a week of enrolment) antimalarial treatment, including two individuals, one infectious and one non-infectious, with Plasmodium ovale co-infection. In Kilifi, five individuals had positive RDT results (parasite densities by microscopy, range 1,240 – 21,500 parasites per μ L) and body temperatures higher than 37.5°C; one of these individuals infected mosquitoes in feeding assays, the others participated in feeding experiments but were non-infectious. Data on recent or current malaria symptoms were not collected for Mbita participants. Our observation that gametocyte carriage is common in asymptomatic infections, 44.2 – 90.0% by Pfs25 mRNA QT-NASBA in those individuals with 18S qPCR positive results in our surveys, together with evidence from a previous study in Mbita where asymptomatic gametocytaemic children were substantially more infectious than symptomatic gametocytaemic children³⁰, suggest that individuals without malaria symptoms are likely to contribute more to local transmission compared to symptomatic individuals. Acute symptomatic infections with high parasite densities are likely to be characterised by a shorter duration of infection and thus shorter time-window that allows gametocyte production compared to infections that are more

*chronic in nature*³¹. *Despite these considerations that support the importance of asymptomatic infections for the human infectious reservoir, it is of great relevance to directly compare onward transmission from symptomatic and asymptomatic infections in the same setting. Such studies require a design where participants are recruited from both local health facilities and the communities they serve.* ~~previous study in Mbita where asymptomatic gametocytaemic children were substantially more infectious than symptomatic gametocytaemic children~~³⁰, ~~these observations indicate that the contribution of individuals developing malaria symptoms to transmission might be largely overwhelmed by the contribution of those infected without symptoms, especially during dry seasons.~~”

Reviewer's comment

Minor comments:

1. It might be worth noting in the discussion that 18S qPCR detected all infectious individuals, arguing against the need for ultrasensitive assays outside of research studies.

Answer to reviewer's comment

Correct – in our study settings, 18S qPCR detected falciparum parasites in all infectious individuals, which suggests that this assay might be a useful tool to exclude non-infectious individuals. However, it is currently unclear what the kinetics of parasite densities are in chronic submicroscopic infections. It is conceivable that some infections may only be detectable by ultra-sensitive methods at one time-point to increase in density and likelihood of transmission in the future; this point has now been made in the *Discussion* section (see below). Whilst 18S qPCR detects all infectious individuals, more user-friendly tools such as a more sensitive RDT may be desirable (see answer to Reviewer #2 comment).

*“One of the strengths of the current study was that we did not select individuals based on parasite status. Prior screening by molecular assays may have increased the proportion of study participants that was infectious to mosquitoes but would have left uncertainties about the transmission potential of undetected infections^{6,20}. We therefore recruited participants for feeding assays from the general population and successfully used molecular diagnostics in 3 of 4 study sites. In our surveys, all infectious individuals with molecular assays results available had parasites detected by 18S qPCR and Pfs25 mRNA QT-NASBA, except one infectious individual believed to have transmitted *P. malariae* parasites. This suggests that these assays might be useful to exclude non-infectious individuals. However, it is currently unclear what the kinetics of parasite densities are in chronic submicroscopic infections and conceivable that some infections that are not detectable by these sensitive assays at one time-point may increase in density and likelihood of transmission in the future.”*

Reviewer’s comment

2. Line 252 – sentence not finished.

Answer to reviewer’s comment

Corrected

“In Laye and Mbita, children younger than 15 years represented the majority of infectious individuals during dry season (76.7 and 65.8%, respectively), whereas during wet season, ~~although~~ adults and children contributed similar numbers of infectious individuals to the local infectious reservoir.”

Reviewer’s comment

3. Lines 360-363 – sentence fragment.

Answer to reviewer's comment

We have now modified the following sentences based on this and other comments (see answers to Reviewer #2's comments):

“In our two surveys in Burkina Faso, parasite densities below 100 parasites per μL were detected by research quality microscopy in 35.2 and 41.5% of infectious individuals, who were responsible for 45.4 and 67.2% of infected mosquitoes (Figure 4), suggesting that a non-negligible proportion of transmission events may be missed by routine microscopy but not necessarily by research microscopy where a larger number of microscopic fields are screened (200 – 400 fields in our study). Both parasite quantification by microscopy and qPCR have limitations and ultimately the detectability of the infectious reservoir may need to be judged against diagnostic practices that are relevant to guide interventions in the field. If transmissible low-density infections could be targeted by interventions using improved diagnostics, such as highly sensitive RDTs, or that include individuals irrespective of parasite status, transmission might be reduced more effectively and rapidly.”

Reviewer #2 (Remarks to the Author):

Reviewer's comment

This is a very well-written paper on a novel analysis that is of great relevance to the field. I strongly support its publication.

Below please find suggestions that I think will increase the impact of the paper on its audience.

- L87 The wording should be modified here. "Conflicting indications" suggests a lack of reproducibility between previous experimental results. But the two previous citations have very different sampling frames -- cross-sectional (Burkina) vs. uncomplicated malaria incidents (Cambodia) -- which results in different distributions of gametocyte density and possibly also different relationships between density and infectiousness.

Answer to reviewer's comment

We agree with the reviewer that these studies are not necessarily comparable and have now modified this sentence to mention that the Cambodian study only included clinical cases and was not a random survey:

“Only one of these studies, conducted in an area of intense malaria transmission in Burkina Faso, concurrently quantified asexual stage parasites and the transmissible sexual stage parasites (gametocytes) by molecular assays and concluded that up to 17% of mosquito infections are caused by submicroscopic parasite carriage in humans¹². In contrast, in Cambodia a hospital-based study involving uncomplicated malaria cases reported much lower infectivity of submicroscopic infections¹³. These ~~contrasting~~ findings of these different studies highlight the need for more extensive assessments of infectivity to mosquitoes across a range of endemicities.”

Reviewer's comment

- L89 Related to the above comment, the "need for more assessments of infectivity" should consider not just "a range of endemicities" but also a sufficient quantification of parasite and gametocyte densities (and other factors) to inform a coherent

understanding of the mechanisms responsible for variation in human-to-mosquito infectiousness.

Answer to reviewer's comment

Thank you. We have incorporated the suggestion in the following sentences:

“~~These contrasting findings~~ of these different studies highlight the need for more extensive assessments of infectivity to mosquitoes across a range of endemicities. Such xenodiagnostic surveys select individuals regardless of parasite carriage, incorporate concurrent assessments of parasite and gametocyte densities by molecular assays, which could provide insights into the mechanisms responsible for the variation in human-to-mosquito infectiousness, and take into account that individuals are not equally bitten by mosquitoes and consequently have different numbers of opportunities to transmit¹⁴.”

Reviewer's comment

- L105 What months do the dry and wet season samplings correspond to? This doesn't appear in Methods either (L457)? Although it is stated for the mosquito sampling (L533).

Answer to reviewer's comment

The months when dry and wet season infectivity surveys were performed are included in Table 1. We refer to this table in the first sentence of the *Results* section.

Reviewer's comment

- L126 Supplemental Figure 3 is very useful and might be cited already in the discussion of Figure 1b.

Answer to reviewer's comment

We have modified the *Results* section to refer to Figure S3 (now Figure S1) in the discussion of Figure 1b:

“The detectability of infections by microscopy was associated with qPCR parasite density (Figure S1); the median (interquartile range [IQR]) parasite densities estimated by 18S qPCR in microscopically subpatent and patent infections were 4.9 (1.0 – 32.6) and 387.3 (95.1 – 1244.5) parasites per μ L, respectively.”

Reviewer's comment

- L133 It is hard to see how the 139 entries in the Kilifi (dry) distribution (Fig 1b), especially the lowest few bins with ~10 entries each can abruptly drop to 0 without a different threshold being applied to that data. Did I miss something in the Methods (L501 vs. L504)?

Answer to reviewer's comment

In Figure 1b, the y-axes represent the proportions of individuals with different parasite levels quantified by 18S qPCR; individuals with no parasites detected by 18S qPCR are not included in this figure. By showing proportions rather than absolute numbers we aimed to ensure that the graphs of the different surveys, that had different numbers of infected individuals, are comparable. Only 52 individuals carried parasites by 18S qPCR during the dry season in Kilifi; 11 of these 52 had parasites levels below 1 parasite per μ L. Since in Figure 1b

densities between 0.01 (the lower limit of detection) and 1 parasite per μL are grouped in 6 bins, we believe this drop noted by the reviewer is related to low absolute numbers of positive samples during that particular survey. We have clarified this in a modified legend to mention the numbers of observations included in Figure 1b:

“Figure 1. Age-specific asexual stage parasites prevalence by microscopy (a) and distributions of parasite densities (b) by study site. In a, 95% exact binomial confidence intervals are presented for microscopy-based parasite prevalence estimates. In b, parasite densities quantified by DNA-based 18S qPCR are presented (\log_{10} scale) for both patent and subpatent infections. This panel only includes 18S qPCR-positive samples (100, 166, 52 and 105 in Burkina Faso dry and wet season surveys, and Kilifi dry and wet season surveys, respectively).”

Reviewer’s comment

- L116 Some statement on the expected case-management rates in the different sites and their differential impact on parasite prevalence at different seasons depending on endemicity (Fig 1a) would be appreciated.

Answer to reviewer’s comment

Based on comments from reviewer 1, we have expanded our section on the transmission potential from clinical malaria cases. All study sites in Burkina Faso and Kenya were located within walking distance from healthcare facilities where individuals with clinical malaria symptoms have access to diagnosis and antimalarial treatment. We did not specifically monitor treatment rates or, perhaps equally important, treatment adherence rates. Additionally, no active surveillance of cases or community chemotherapy campaigns were undertaken in the study sites during our work although seasonal malaria chemoprophylaxis for children under 5 years of age has recently been adopted in Burkina Faso (i.e. after the completion of data collection for the current study). We have now modified the *Methods* section:

“Our surveys were conducted in an area of intense malaria transmission in Burkina Faso and areas of low and moderate malaria transmission in Kenya. In Burkina Faso, the dry season survey was performed in the village of Laye and the wet season survey in nearby Balonghin. Both villages are characterized by intense seasonal transmission^{12,31}. In Kenya, dry and wet season surveys were performed in Mbita, on the shores of Lake Victoria in the Suba District in Western Kenya, an area characterised by moderate malaria transmission intensity³². In Kilifi, in coastal Kenya, malaria transmission occurs throughout the year, but peaks during the wet season³³. Over the last 25 years, malaria incidence in this area has declined to very low levels³⁴. Of note, health facilities, clinics, were available in all study villages, and artemisinin-based combination therapy is used as first-line treatment of clinical cases in all sites. At the time of these surveys, seasonal malaria chemoprophylaxis for children under 5 years of age was not part of national guidelines in Burkina Faso.”

Reviewer’s comment

- L174/L389 The discussion of age-specific ITN usage is appreciated, but can you also comment on any differences in mosquito species collected between Mbita and Balonghin? Are there any differences in household composition? Is it not possible to present these results (Fig 2) in categories of, e.g. the largest household member, another adult, the largest child, etc.? It is not clear how the dominant feature of Fig 2b, household-to-household variability, interacts with mosquito feeding choices and the observed age patterns.

Answer to reviewer’s comment

We appreciated these comments on the field collection of mosquitoes that was used in the current manuscript to quantify mosquito exposure (as we argue an essential component in transmission potential). Those data are currently being analysed for a more in-depth description of feeding rates, insecticide resistance and sporozoite rates (Guelbeogo *et al.*, in preparation). *Anopheles gambiae sensu stricto* and *A. coluzzii* were the most prevalence

species in Balonghin (and our currently ongoing speciation data confirm this), while previous studies (Mathenge et al. *Comparative field evaluation of the Mbita trap, the Centers for Disease Control light trap, and the human landing catch for sampling of malaria vectors in western Kenya*. Am J Trop Med Hyg 2004) have shown that in Mbita *A. funestus* is also an important vector. While degree of anthropophily is known to vary among *Anopheles* species, our aim was simply to quantify natural sampling by vectors to adjust our estimates of the infectious reservoir (for example, see Port and Boreham. *The relationship of host size to feeding by mosquitoes of the Anopheles gambiae Giles complex*, 1980; in this reference, a table is included that describes results of other studies with similar goals). Our results suggest more trapping is required to assess species specific differences. We have now mentioned in the *Discussion* section that the differences in age-specific exposure in Mbita versus Balonghin could be related to vector composition:

“The difference in mosquito feeding choices between the sites may be partially explained by differences in bed net use; in Mbita, adults reportedly slept under nets more often (87.7%) than children < 5 years (77.8%) and schoolchildren (63.8%), while in Burkina Faso, reported bed net usage was similar in all age groups (89.5, 87.1 and 88.9% for young children, schoolchildren and adults). Differences in species composition of local vector populations is another possible explanation, although evidence for between-Anopheles species variation in feeding behaviour with regards to human hosts at the individual level is limited²⁶. Although we also observed considerable variation in mosquito exposure between households, and ideally we would have been able to quantify mosquito exposure for every xenodiagnostic survey participant, our observations of mosquito feeding preferences in a selection of households allow a better interpretation of mosquito feeding experiments by extrapolating from standardised numbers of mosquitoes in feeding assays to actual mosquito sampling rates that are strongly age-dependent. When adjusting our estimates for mosquito exposure, the relative contribution of adults to the infectious reservoir nearly doubles in Burkina Faso and resulted in a more modest increase in the contribution of older individuals to the infectious reservoir in our Kenya sites.”

Although in some study households no bloodfed mosquitoes were matched to residents, we believe our analysis is appropriate for the current purposes: obtaining an age-specific

mosquito exposure estimate, that truly seems to be predominantly age-driven in Burkina Faso. By using mixed effects models that have a random intercept at the household level, we are accounting for household-to-household variability in mosquito abundance in this analysis. We have included this information in the *Methods* section:

“Initially, we fit a model that included data from both study sites and tested for interaction between age effects and study site effects. Since there was a significant interaction ($P = 0.001$) between study site and age effects, we present separate analyses for the different sites. For the entomological study in Balonghin, this analysis was adjusted for survey (fixed effect). To account for data correlatedness and household-level differences in vector abundance, the models for both sites had household as random effect.”

In Figure 2d, we present the relationship between mosquito exposure and age as a continuous variable. To address the reviewer’s question, we created an alternative version of this figure that only includes data from households where at least one mosquito was matched to house occupants (see below). In this new graph, the same relationship between age and mosquito exposure as seen in Figure 2d is observed.

Our presentation of the age patterns in mosquito exposure are therefore correct, although obviously there is considerable variation between households and therefore between individuals in the same age category. The current data with single time-point assessments of infectivity in the surveys in Balonghin and Laye and mosquito exposure data for a selection of households in Balonghin only do not allow us to estimate individual-level mosquito exposure of those individuals for whom we have infectiousness data. Had mosquito exposure data been available for every xenodiagnostic survey participant, we would be able to estimate individual-specific number of secondary mosquito infections rather than average numbers by age group. To mention this unavoidable shortcoming, we revised the discussion section:

‘Although we also observed considerable variation in mosquito exposure between households, and ideally we would have been able to quantify mosquito exposure for every xenodiagnostic survey participant, our observations of mosquito feeding preferences in a selection of households allow a better interpretation of mosquito feeding experiments by extrapolating from standardised numbers of mosquitoes in feeding assays to actual mosquito sampling rates that are strongly age-dependent. When adjusting our estimates for mosquito exposure, the relative contribution of adults to the infectious reservoir nearly doubles in

Burkina Faso and resulted in a more modest increase in the contribution of older individuals to the infectious reservoir in our Kenya sites.

Reviewer's comment

- L201/L567 In the several hundred mixed bloodmeals, does the distribution of number of loci with multiple alleles (e.g. 3/10 vs. 10/10) give some indication on the likelihood that the multiple feeds were on members of the same nuclear family within the same household? Regardless, this distribution would be an interesting addition to the supplement.

Answer to reviewer's comment

We agree with the reviewer that these data are a valuable addition to the manuscript. Mosquito blood meals with three or more alleles in most loci tested are, in theory, likely to have as different sources of human DNA individuals from different families, since related members of the same nuclear family are more likely to share their alleles. To confirm whether this is the case, unambiguous matching of multiple source blood meals to study participants would be necessary. This is generally not possible. We agree with the reviewer that it would be interesting to present the distribution of these multiple source blood meals by number of loci with multiple alleles. We have now included the table below in the *Supplementary Information file* (Table S4).

Table S4. Distribution of multiple source blood meals by number of loci with multiple alleles (three or more alleles per loci)

Number of loci with multiple alleles (3 or more)	Study sites	
	Balonghin	Mbita
3	45	27
4	29	13
5	38	21
6	28	8
7	11	12
8	2	2
9	0	4
10	0	0

Additionally, we have included in the *Results* section and in the *Supplementary Information* file the results of a sensitivity analysis that allocates multiple-source blood meals to those individuals with lowest allelic distances to the genotypes present in these blood meals; this analysis corroborates the results of the model that excluded multiple source meals.

“To further assess the sensitivity of our outcomes to biting preferences of mosquitoes with multiple human blood sources, we re-analysed our data assuming 1) that multiple-source meals were single blood meals equally allocated to study participants living in the household where they were collected, or 2) that individuals who had the lowest genetic distances to the genotypes present in these meals were blood sources: under these different assumptions, adults were still seven to eleven times more likely to be bitten by mosquitoes than young children in Balonghin (see Supplementary Information).”

Reviewer’s comment

- L243 More information on the very high oocyst-count infection would be useful (Fig 3C), e.g. what is the gametocyte density measured by microscopy. Also the 197-oocyst infection in Mbita. Perhaps two more columns in Table S1 (microscopy asexual and gametocyte densities)?

Answer to reviewer's comment

The gametocyte density measured by microscopy of the study participant who infected the mosquito with 71 oocysts was 572 gametocytes per μL ; the gametocyte density of the participant from Mbita who infected a mosquito with 197 oocysts was 23 gametocytes per μL . To address the reviewer's comment, we have now added two columns in Table S1 that show gametocyte densities by *Pfs25* mRNA QT-NASBA and medians and ranges of oocyst counts.

Study Site	Survey	Proportion (n/N) infected mosquitoes	Median (range) oocyst count	Patent falciparum asexual	Patent falciparum gametocytes	P. malariae parasites	P. ovale parasites	Pfs25 mRNA QT-NASBA	18S rRNA QT-NASBA	18S qPCR	18S qPCR (parasites per μ L)	Pfs25 QT-NASBA (gametocytes per μ L)	Molecular confirmation of falciparum mosquito infection
Laye	Dry	0.03 (2/77)	(1, 1)	Negative	Negative	Positive	Negative	Negative	Negative	Negative	0.0	0.0	Negative
Laye	Dry	0.10 (7/71)	3 (1–4)	Negative	Positive	Negative	Negative	Positive	Positive	Positive	0.4	110.1	All P. falciparum
Laye	Dry	0.27 (21/78)	2 (1–5)	Negative	Positive	Negative	Negative	Positive	Positive	Positive	0.7	9.2	All P. falciparum
Laye	Dry	0.04 (3/73)	1 (1–1)	Positive	Positive	Negative	Negative	Positive	Positive	Positive	1.0	7.2	All P. falciparum
Laye	Dry	0.05 (3/65)	1 (1–1)	Negative	Positive	Negative	Negative	Positive	Negative	Positive	8.6	10.8	All P. falciparum
Laye	Dry	0.16 (16/98)	1 (1–5)	Negative	Negative	Negative	Negative	Positive	Positive	Positive	11.9	10.9	All P. falciparum
Laye	Dry	0.02 (2/96)	(1, 3)	Positive	Negative	Negative	Negative	Positive	Positive	Positive	150.3	50.3	All P. falciparum
Laye	Dry	0.20 (14/71)	3.5 (1–11)	Positive	Negative	Negative	Negative	Positive	Positive	Positive	200.9	88.1	All P. falciparum
Laye	Dry	0.28 (24/87)	2.5 (1–22)	Positive	Positive	Negative	Negative	Positive	Positive	Positive	218.8	42.5	All P. falciparum
Laye	Dry	0.03 (3/88)	2 (1–2)	Positive	Positive	Negative	Negative	Positive	Positive	Positive	281.2	11.0	All P. falciparum
Laye	Dry	0.01 (1/97)	1	Positive	Negative	Negative	Negative	Positive	Positive	Positive	399.0	24.1	All P. falciparum
Laye	Dry	0.02 (2/81)	(1, 1)	Positive	Negative	Negative	Negative	Positive	Positive	Positive	431.5	13.0	All P. falciparum
Laye	Dry	0.01 (1/76)	1	Positive	Negative	Negative	Negative	Positive	Positive	Positive	1119.4	21.4	All P. falciparum
Laye	Dry	0.13 (11/84)	2 (1–4)	Positive	Negative	Negative	Negative	Positive	Negative	Positive	1782.4	54.3	All P. falciparum
Balonghin	Wet	0.03 (1/39)	1	Negative	Positive	Negative	Negative	Positive	Positive	Positive	0.4	21.2	All P. falciparum
Balonghin	Wet	0.05 (2/39)	(1, 2)	Negative	Positive	Negative	Negative	Positive	Positive	Positive	1.3	116.9	All P. falciparum
Balonghin	Wet	0.33 (14/43)	2 (1–5)	Positive	Positive	Negative	Negative	Positive	Positive	Positive	15.1	106.2	All P. falciparum
Balonghin	Wet	0.05 (2/44)	(2, 2)	Positive	Negative	Negative	Negative	Positive	Positive	Positive	54.1	68.1	All P. falciparum
Balonghin	Wet	0.14 (5/35)	2 (1–5)	Positive	Positive	Negative	Negative	Positive	Positive	Positive	114.3	89.9	All P. falciparum
Balonghin	Wet	0.03 (1/33)	1	Positive	Positive	Negative	Negative	Positive	Positive	Positive	356.5	32.7	All P. falciparum
Balonghin	Wet	0.34 (17/50)	3 (1–16)	Positive	Negative	Negative	Negative	Positive	Positive	Positive	435.5	161.1	All P. falciparum
Balonghin	Wet	0.29 (15/51)	1 (1–15)	Positive	Positive	Negative	Negative	Positive	Positive	Positive	450.8	144.5	All P. falciparum
Balonghin	Wet	0.49 (17/35)	4 (1–15)	Positive	Positive	Negative	Positive	Positive	Positive	Positive	712.9	0.04	Negative
Balonghin	Wet	0.23 (8/35)	4 (2–13)	Positive	Positive	Negative	Negative	Positive	Positive	Positive	1324.3	162.5	All P. falciparum
Balonghin	Wet	0.27 (12/45)	4 (1–9)	Positive	Positive	Negative	Negative	Positive	Positive	Positive	2264.6	220.8	All P. falciparum
Balonghin	Wet	0.50 (24/48)	16 (1–71)	Positive	Positive	Negative	Negative	Positive	Positive	Positive	4305.3	3228.5	All P. falciparum
Balonghin	Wet	0.07 (3/41)	8 (3–8)	Positive	Positive	Negative	Negative	Positive	Positive	Positive	30478.9	144.9	All P. falciparum
Mbita	Dry	0.07 (4/60)	1 (1–1)	Negative	Negative	Negative	Negative	-	-	-	-	-	1 mosq. P. falciparum
Mbita	Dry	0.02 (1/56)	1	Positive	Negative	Positive	Negative	-	-	-	-	-	-
Mbita	Dry	0.57 (17/30)	8 (2–197)	Negative	Positive	Positive	Negative	-	-	-	-	-	1 mosq. P. malariae
Mbita	Dry	0.02 (1/44)	3	Positive	Positive	Positive	Negative	-	-	-	-	-	-
Mbita	Dry	0.02 (1/60)	1	Positive	Positive	Negative	Negative	-	-	-	-	-	-
Mbita	Dry	0.02 (1/60)	1	Positive	Positive	Negative	Negative	-	-	-	-	-	-
Mbita	Dry	0.05 (3/60)	1 (1–1)	Positive	Positive	Positive	Negative	-	-	-	-	-	1 mosq. P. malariae
Mbita	Wet	0.07 (2/30)	(1, 2)	Negative	Negative	Positive	Negative	-	-	-	-	-	-
Mbita	Wet	0.08 (3/37)	2 (1–4)	Positive	Positive	Positive	Negative	-	-	-	-	-	-
Kilifi	Wet	0.07 (1/15)	1	Negative	Negative	Negative	Negative	Positive	Positive	Positive	30.7	2.0	Positive
Kilifi	Wet	0.04 (2/45)	(1, 4)	Positive	Positive	Negative	Negative	Positive	Positive	Positive	1361.4	266.1	Positive
Kilifi	Wet	0.03 (1/32)	3	Positive	Negative	Negative	Negative	Positive	Positive	Positive	5345.8	107.9	Positive

Reviewer's comment

- Fig 3b would be more informative if it weren't so squished. Might I recommend coloring by microscopic gametocytemia (the current coloring is redundant); stretching horizontally; and changing to the y-axis to a log-scale down to the lowest non-zero value, then an axis break, then a zero bin.

Answer to reviewer's comment

We are grateful for the reviewer's suggestions. We have now modified this figure (see below) and its legend. In this version, green circles represent individuals with patent gametocytes.

“...In b, the proportion of mosquitoes infected in individual feeding experiments (y axis) and gametocytes densities (x axis) are shown. Data from all surveys are presented: gametocytes densities were quantified by Pfs25 mRNA QT-NASBA in samples collected in Burkina Faso and Kilifi, and by microscopy for Mbita participants. Green circles correspond to samples with patent gametocytes. Both the x axis and the segment of the y axis that ranges from 0.01 to 1 are in \log_{10} -scale. Individuals who did not infect mosquitoes are presented in a separate segment of y axis that only includes the 0 y-coordinate.

...”

Reviewer’s comment

- L290 Given the large fraction of infections arising from 10-100/uL microscopic density infections, a mention of high-sensitivity RDTs might be relevant here.

Answer to reviewer’s comment

We have included the following sentence in the *Discussion* section (see also Reviewer #1 comment):

“If transmissible low-density infections could be targeted by interventions using improved diagnostics, such as highly sensitive RDTs, or that include individuals irrespective of parasite status, transmission might be reduced more effectively and rapidly.”

Reviewer’s comment

- Fig 4 The title of Figure 4 should probably be something more like "Proportion of infected mosquitoes by parasite density". The reader should not have to jump to Table 2 to get a sense of the significance of these results. Adding N_humans=14,

N_mosquitoes=110 directly on the figure for Burkina Faso dry season, etc. would be an improvement.

Answer to reviewer's comment

We agree with the reviewer that the title “Proportion of infected mosquitoes by parasite density” more clearly summarises the results presented this figure. We have now modified the title and included in the legend the numbers of infectious individuals and infected mosquitoes in each survey to facilitate interpretation by the reader. Please note that these numbers do not include infectious individuals with non-falciparum malaria,

*“Figure 4. Proportion of infected mosquitoes by parasite density. Age-specific prevalences of falciparum malaria parasites by microscopy and PCR and infectiousness prevalences by microscopy-defined parasite density were used to estimate the proportions of *P. falciparum* infected mosquitoes in each community; demographic age structure in Sub-Saharan Africa populations was used to standardise estimates. Individuals with evidence of non-falciparum malaria infections were excluded (N=2). The top panels represent the contributions of human infections with different parasite densities to local mosquito infections, after adjusting for population age structure and age-and-parasite density-specific probabilities of mosquito infection in feeding assays; in the bottom panels, age-specific relative mosquito exposure data were used. These calculations were based on 13, 12 and 3 infectious individuals and 108, 104 and 4 infected mosquitoes in the Burkina Faso dry and wet season surveys and in the Kilifi wet season survey, respectively. Data from Mbita are not presented as most infectious individuals in this setting had *P. malariae* co-infections.*

Reviewer's comment

- L296 (and elsewhere) Please be careful to state clearly what densities (asexual, gametocyte, microscopy, NASBA) are being used in different places. In this example,

the specifics are in Methods L584-586 but that leaves the reader guessing what is being shown in the results and Figure 4.

Answer to reviewer's comment

The densities mentioned in line 296 refer to microscopy-based parasite quantification. We have now modified this sentence to clarify which method was used in this analysis. In Figure 4, microscopy was used to categorise individuals according to their parasite levels; this is stated in Figure 4 legend.

*“Before adjusting for mosquito exposure, 19.6 – 52.1% of *P. falciparum* infected mosquitoes became infected from individuals with *P. falciparum* parasite densities below 100 parasites per μL by microscopy, including individuals who had subpatent parasites detectable by PCR (Figure 4, top panels). After adjusting for mosquito exposure, these percentages increased to 44.6 – 76.6% (Figure 4, bottom panels).”*

Reviewer's comment

- L324 "multi-site" here is being used to contrast with previous work done in multiple sites separated by a few 10s of kilometers in Burkina Faso [Ref 12]. Multi-country or -region might be more clear?

Answer to reviewer's comment

We agree that “multi-region” would be more appropriate. We have now modified the following sentences in the *Abstract* and *Discussion* section:

“We report on the first multi-region study to assess population-wide malaria transmission potential based on 1,209 mosquito feeding assays in endemic areas in Burkina Faso and Kenya.”

“Here, we report the first multi-region assessment of malaria transmission using a standardised mosquito feeding protocol and highly sensitive molecular assays for parasite and gametocyte quantification.”

Reviewer’s comment

- L339 The justification of feeding without screening is acceptable here. But before L343 (“There is accumulating evidence...”) one has to finish this line of reasoning and make clear that 0/30 infectious humans were 18S- and 25S-, although in Mbita we can't say. Then move on to a new paragraph related to subpatent infections.

Answer to reviewer’s comment

We appreciate this suggestion and have now modified this paragraph:

“One of the strengths of the current study was that we did not select individuals based on parasite status. Prior screening by molecular assays may have increased the proportion of study participants that was infectious to mosquitoes but would have left uncertainties about the transmission potential of undetected infections^{6,20}. We therefore recruited participants for feeding assays from the general population and successfully used molecular diagnostics in 3 of 4 study sites. In our surveys, all infectious individuals with molecular assays results available had parasites detected by 18S qPCR and Pfs25 mRNA QT-NASBA, except one infectious individual believed to have transmitted P. malariae parasites. This suggests that these assays might be useful to exclude non-infectious individuals. However, it is currently unclear what the kinetics of parasite densities are in chronic submicroscopic infections and

conceivable that some infections that are not detectable by these sensitive assays at one time-point may increase in density and likelihood of transmission in the future.

There is accumulating evidence that in all endemicities substantial proportions of falciparum infections are subpatent, i.e. below the limit of detection of conventional field diagnostics⁷.

...”

Reviewer’s comment

- L347 To strengthen this point, it might be worth mentioning that the SE Asian findings involved both fitting the low end of the distribution and imputing the Pf/Pv allocation of unspiciated low density samples based on the speciated ratio.

Answer to reviewer’s comment

We have now modified this sentence to provide more information about the study by Imwong and colleagues:

“In contrast to findings with high-volume qPCR from a large epidemiological study in Southeast Asia⁶, where the percentage of undetectable infections was estimated based on distributions of quantifiable parasite densities, we found no evidence for a significant number of infections being missed by 18S qPCR, as indicated by Figure 1b.”

Reviewer’s comment

- L355 This sentence is a bit of a non sequitur. If you feel it's an important point, it needs to be followed with a reference to Fig S3 and a discussion of different

contributions to measurement uncertainty (variable white blood cell counts, log-normal errors from amplification, etc.)

Answer to reviewer's comment

We agree with the reviewer that the statement comparing 18S qPCR densities and microscopy-defined densities would require additional discussion on the factors that influence parasite quantification by microscopy (e.g., variability in white blood cell counts) and molecular methods (e.g., extraction and amplification efficiency) and decided to remove this sentence as it is not a major result of this manuscript. However, we believe that the comparison between research quality microscopy and routine microscopy is a valid one, as they share the same sources of variability and the main difference between them is presumably the number of white blood cells counted to determine absence of infection. The paragraph mentioned by the reviewer is now:

“There is accumulating evidence that in all endemicities substantial proportions of falciparum infections are subpatent, i.e. below the limit of detection of conventional field diagnostics⁷. In line with this, we detected a considerably larger number of infections with molecular assays than microscopy. In contrast to findings with high-volume qPCR from a large epidemiological study in Southeast Asia⁶, where the percentage of undetectable infections was estimated based on distributions of quantifiable parasite densities, we found no evidence for a significant number of infections being missed by 18S qPCR, as indicated by Figure 1b. There is considerable interest in quantifying the contribution of low density, submicroscopic, infections to onward transmission. In practice, this definition is influenced by the sensitivity of microscopy and molecular assays, both of which can vary substantially between settings. Whilst a lower limit of detection of routine microscopy is assumed to be 50–100 parasites per μL 21, 25.5% (62/245) of infections with 18S qPCR densities below 100 parasites per μL were identified by research quality microscopy that involved screening of 200–400 fields. In our two surveys in Burkina Faso, parasite densities below 100 parasites per μL were detected by research quality microscopy in 35.2 and 41.5% of infectious individuals, who were responsible for 45.4 and 67.2% of infected mosquitoes (Figure 4),

suggesting that a non-negligible proportion of transmission events may be missed by routine microscopy but not necessarily by research microscopy where a larger number of microscopic fields are screened (200 – 400 fields in our study). Both parasite quantification by microscopy and qPCR have limitations and ultimately the detectability of the infectious reservoir may need to be judged against diagnostic practices that are relevant to guide interventions in the field. If transmissible low-density infections could be targeted by interventions using improved diagnostics, such as highly sensitive RDTs, or that include individuals irrespective of parasite status, transmission might be reduced more effectively and rapidly. Of note, in Kilifi, one individual of three who were infectious in feeding experiments did not carry patent parasites. Whilst we believe the low proportion of infectious individuals accurately reflects the low likelihood of transmission in this setting, numbers are limited to draw conclusions on the performance of different diagnostics to identify the human infectious reservoir for malaria. For this, the methodology for xenodiagnostic studies may need to be refined to include sensitive screening tools to identify potentially infectious individuals in low transmission areas and provide more robust estimates of population infectiousness.

Reviewer's comment

- L357 (related to earlier comment) If you keep the 100/uL qPCR density sentence that precedes this, it should be reinforced that the next sentence is referring to 100/uL by microscopy (greater of asexual and gametocytes as per Methods).

Answer to reviewer's comment

We have now modified the sentence to explicitly state that we are referring to microscopy-based quantification (see answer to previous comment).

Reviewer's comment

- L370 Greater than 10/uL by what detection method?

Answer to reviewer's comment

In our study, most infectious individuals had densities of 10 or more gametocytes per μL , as quantified by *Pfs25* mRNA QT-NASBA. We have now modified the sentence to which the reviewer refers:

*“In our surveys, mosquito infection rates were loosely associated with gametocyte densities^{20,22} and most infectious individuals had an estimated density of 10 or more gametocytes per μL by *Pfs25* mRNA QT-NASBA.”*

Reviewer's comment

- L570 Are the houses where individuals declined to participate in the human blood typing (15% in Mbita) included in the numbers of unmatched bloodfed mosquitoes?

Answer to reviewer's comment

Yes, data from the fifteen houses where at least one individual did not provide blood sample were included in the analysis. In Mbita, 174 mosquito blood meals were not matched to individuals living in the same house where they were collected. Whilst 9 of these blood meals matched study participants living in a different study household, 165/174 were not matched to any study participant. 81.8% of these unmatched bloodfed mosquitoes came from houses with at least one individual who was not sampled. We have now mentioned, in the *Discussion* section, that this is a likely explanation of the higher number of unmatched mosquitoes in Mbita compared to Balonghin (see paragraph below). Importantly, as with any

epidemiological study, unless non-participation (and no blood sample) was associated with both age and exposure to mosquitoes, it should not bias our age-related mosquito exposure estimates.

“A number of mosquito blood meals (14.4% in Balonghin and 25.2% in Mbita) could not be linked to residents of study houses. In Balonghin, nearly all household occupants provided a blood sample that allowed genetic matching to mosquito blood meals and this suggests indoor resting of mosquitoes that fed elsewhere. In Mbita however, the higher percentage of unmatched mosquitoes could be at least partially explained by the fact that 13.3% (25/188) of household occupants did not provide blood samples for matching.”

Reviewer’s comment

There are a few punctuation issues and awkward sentences in the text. A few examples below:

- L64 "infectious diseases, and ..."
- L73 "The premise being..." (fragment)
- L338 "from areas" "in areas" ??

Answer to reviewer’s comment

Thank you. We have now corrected these and other issues:

“Heterogeneity in the transmission potential of individual hosts is a common feature of many infectious diseases, and the identification of individuals who disproportionately contribute to onward transmission has attracted much attention.”

“Initiatives to further reduce the burden of malaria, as well as efforts to contain the spread of artemisinin resistant malaria parasites in Southeast Asia⁵, require a thorough understanding of the human infectious reservoir for malaria, which would allow interventions to be targeted to individuals who are most important for the transmission of infection to mosquitoes.”

“This is broadly in line with the limited data available on the human infectious reservoir¹⁴ that are almost exclusively from areas of intense malaria transmission...”

REVIEWERS' COMMENTS:

Reviewer #1 (Remarks to the Author):

The authors have responded well to all of the criticisms. The paper can be published as is.

Reviewer #2 (Remarks to the Author):

The authors have adequately addressed the comments in their response. I would recommend publication.

Reviewer #1 (Remarks to the Author):

The authors have responded well to all of the criticisms. The paper can be published as is.

Reviewer #2 (Remarks to the Author):

The authors have adequately addressed the comments in their response. I would recommend publication.

Answer to reviewers' comments

We would like to thank both reviewers for their initial comments, which considerably improved this manuscript.